# A mycobacterial effector promotes ferroptosis-dependent pathogenicity and dissemination

Lihua Qiang[1,2,5], Yong Zhang [1,5], Zehui Lei[1,2,5], Zhe Lu[1,2], Shasha Tan[1], Pupu Ge[1], Qiyao Chai[1], Mengyuan Zhao[1,2], Xinwen Zhang[1,2], Bingxi Li[1], Yu Pang[3], Lingqiang Zhang [4] ✉, Cui Hua Liu [1,2] ✉ & Jing Wang [1] ✉

Ferroptosis is a lipid peroxidation-driven and iron-dependent programmed cell death involved in multiple physical processes and various diseases. Emerging evidence suggests that several pathogens manipulate ferroptosis for their pathogenicity and dissemination, but the underlying molecular mechanisms remain elusive. Here, we identify that protein tyrosine phosphatase A (PtpA), an effector secreted by tuberculosis (TB)-causing pathogen *Mycobacterium tuberculosis* (Mtb), triggers ferroptosis to promote Mtb pathogenicity and dissemination. Mechanistically, PtpA, through its Cys11 site, interacts with host RanGDP to enter host cell nucleus. Then, the nuclear PtpA enhances asymmetric dimethylation of histone H3 arginine 2 (H3R2me2a) via targeting protein arginine methyltransferase 6 (PRMT6), thus inhibiting glutathione peroxidase 4 (GPX4) expression, eventually inducing ferroptosis to promote Mtb pathogenicity and dissemination. Taken together, our findings provide insights into molecular mechanisms of pathogen-induced ferroptosis, indicating a potential TB treatment via blocking Mtb PtpA-host PRMT6 interface to target GPX4-dependent ferroptosis.

Cell death modalities, which are common outcomes during pathogen infection, are either beneficial for host defense against the invading pathogens, or being exploited by pathogens for their pathogenicity and dissemination[1,2]. Ferroptosis, a recently defined form of programmed cell death driven by the disorder of lipid peroxide repair systems, plays critical roles in multiple physical processes or various diseases, including infectious diseases[3–6]. Emerging evidence suggests that various pathogens manipulate ferroptosis for their pathogenicity and dissemination[7–9], but the underlying molecular mechanisms remain elusive.

Tuberculosis (TB), a life-threatening chronic infectious disease caused by *Mycobacterium tuberculosis* (Mtb)[10], induces necrotic lesions to facilitate its pathogenicity and dissemination in host tissues[11]. A recent study shows that Mtb-induced necrosis exhibits important hallmarks of ferroptosis, including reduced levels of glutathione (GSH), glutathione peroxidase 4 (GPX4), as well as accumulation of lipid peroxides, and blocking the host ferroptotic pathway suppresses pathogen dissemination[8]. Moreover, TB patients being treated with anti-ferroptosis drugs, such as vitamin E or selenium enzymes, show improved treatment outcomes[12,13]. But up to now, the regulatory roles and the molecular mechanisms of Mtb effectors involved in host ferroptosis modulation remain largely unexplored, which knowledge could be helpful to provide new strategies and potential targets for TB treatments.

[1]CAS Key Laboratory of Pathogenic Microbiology and Immunology, Institute of Microbiology, Chinese Academy of Sciences, Beijing 100101, China. [2]Savaid Medical School, University of Chinese Academy of Sciences, Beijing 101408, China. [3]Beijing Tuberculosis and Thoracic Tumor Research Institute, Beijing Chest Hospital, Capital Medical University, Beijing 101149, China. [4]State Key Laboratory of Proteomics, National Center for Protein Sciences (Beijing), Beijing Institute of Lifeomics, Beijing 100850, China. [5]These authors contributed equally: Lihua Qiang, Yong Zhang, Zehui Lei. ✉e-mail: zhanglq@nic.bmi.ac.cn; liucuihua@im.ac.cn; wangj6@im.ac.cn

Mtb eukaryotic-like phosphatases and kinases, which play important roles in regulating host cellular physiological processes including apoptosis, cell proliferation, phagosome acidification, autophagy flux, and immune signaling pathway[14–17], have emerged as prominent therapeutic targets for anti-TB drugs, but their regulatory roles and mechanisms in host ferroptosis remain unknown. We thus performed a small-scale screen for Mtb effectors regulating ferroptosis among Mtb eukaryotic-like phosphatases and kinases, and identified protein tyrosine phosphatase A (PtpA) as a pro-ferroptotic effector. Further transcriptome sequencing analysis showed that PtpA inhibited the expression of GSH metabolism-associated genes, especially GPX4, a central repressor of ferroptosis[18]. Mechanistically, PtpA, through its Cys11 site, interacted with RanGDP to enter host cell nucleus. Subsequently, the nuclear PtpA directly interacted with protein arginine methyltransferase 6 (PRMT6) to increase asymmetric dimethylation of histone H3 arginine 2 (H3R2me2a), thereby inhibiting the expression of GPX4, followed by ferroptosis, leading to the promotion of pathogen pathogenicity and dissemination. Together, this study provides insights into the molecular mechanism underlying PtpA-mediated host ferroptosis during Mtb infection, and suggests a potential Mtb-host interface-based TB treatment.

## Results

### Mtb PtpA induces GPX4-mediated ferroptosis
With an aim to explore the regulatory roles of Mtb effectors involved in host ferroptosis, we sought to screen Mtb eukaryotic-like phosphatases and kinases to identify effectors regulating ferroptosis. As reported, Mtb-induced ferroptosis in host cells is characterized by GSH depletion and GPX4 inhibition[8], which could be caused by ferroptosis inducers including Erastin (system $x_c^-$ inhibitor), buthionine sulfoximine (BSO, GSH synthase inhibitor), or RSL3 (GPX4 inhibitor)[4,19]. We thus started by identifying host cells susceptible to ferroptosis as well as specific ferroptosis inducers during Mtb infection, and our results demonstrated that all the examined cells, including peripheral blood mononuclear cells (PBMCs), human monocyte-like U937 cells, and human lung carcinoma A549 cells[20,21] were sensitive to RSL3-induced ferroptosis (Supplementary Fig. 1a–c). Next, we measured cell viability of wild-type (WT) A549 cells and cells overexpressing individual Mtb eukaryotic-like phosphatases or kinases after RSL3 treatment, and found that A549 cells overexpressing PtpA (also termed Rv2234) were more sensitive to RSL3-induced ferroptosis than cells overexpressing other Mtb effectors (Supplementary Fig. 1d, e). Consistently, U937 cells stably expressing PtpA were also vulnerable to RSL3-induced ferroptosis (Fig. 1a, b and Supplementary Movies 1, 2). To further investigate the regulatory role of PtpA in ferroptosis during Mtb infection, we deleted the gene encoding ptpA in Mtb H37Rv (Mtb ΔptpA) and complemented Mtb ΔptpA with WT ptpA (Mtb ΔptpA:ptpA) for infection of U937 cells. Compared with cells infected with Mtb ΔptpA strain, the cells infected with WT Mtb or Mtb ΔptpA:ptpA strain showed decreased cell viability, which phenomenon was abolished by ferroptosis inhibitors ferrostatin-1 (Fer-1) and liproxstatin-1 (Lip-1), two potent lipid peroxidation inhibitors (Fig. 1c, d and Supplementary Fig. 2a–c)[22,23]. In addition, cells infected with WT Mtb strain or Mtb ΔptpA:ptpA strain exhibited significantly higher levels of lipid peroxides (a hallmark of ferroptosis) than cells infected with Mtb ΔptpA strain (Fig. 1e, f and Supplementary Fig. 2d). Collectively, these results imply that Mtb PtpA is a critical pro-ferroptotic effector during mycobacterial infection.

To explore how PtpA promotes host ferroptosis, we performed RNA sequencing (RNA-seq) analysis using U937 cells infected with WT or ΔptpA mycobacteria, and identified 340 differentially expressed genes (accession number GEO: GSE199069) (Supplementary Fig. 2e). Gene set enrichment analysis (GSEA) showed the top 10 enriched pathways including GSH metabolism (Supplementary Fig. 2f). According to the enrichment results, the signature GSH metabolism-related genes were significantly upregulated in cells infected with ΔptpA strain, compared with that in cells infected with WT mycobacteria (Supplementary Fig. 2g). We then verified the expression of those genes in U937 cells infected with WT Mtb, Mtb ΔptpA, or Mtb ΔptpA:ptpA strain, and confirmed that PtpA significantly inhibited the transcription of several ferroptosis-regulating genes including GPX4, Isocitrate dehydrogenase 1 (IDH1), glutamate-cysteine ligase modifier subunit (GCLM), and Microsomal glutathione S-transferase 2 (MGST2) (Fig. 1g and Supplementary Fig. 2h), among which GPX4 is a determining factor in RSL3-induced ferroptosis, the form of ferroptosis regulated by PtpA (Fig. 1a). Consistently, we also confirmed that PtpA suppressed GPX4 protein expression during Mtb infection (Fig. 1h). To further confirm that PtpA-induced ferroptosis is mediated by GPX4, we adopted the Tet-On inducible gene expression system to restore GPX4 gene expression by doxycycline (DOX) in U937 cells. As shown by immunoblot analysis, we confirmed that DOX could indeed restore GPX4 expression in cells infected with WT Mtb or Mtb ΔptpA:ptpA strain to the similar level as that in uninfected cells without DOX treatment (Supplementary Fig. 2i). Then, we analyzed the lipid peroxides levels and cell viability of U937 cells, and found that GPX4 restoration suppresses Mtb-induced ferroptosis (Fig. 1i, j). Taken together, our results indicate that Mtb PtpA induces GPX4-mediated ferroptosis in host cells.

### Mtb PtpA, through a non-canonical RanGDP-binding site, enters host cell nucleus to induce ferroptosis
Mtb secretory effectors usually localize in different subcellular compartments to regulate host cellular processes[24–26]. For example, PtpA is regarded not only as a phosphatase in the cytosol to suppress innate immunity, but also as a DNA-binding factor in host cell nucleus to promote epithelial cell proliferation[27,28]. We first confirmed that PtpA was localized in both cytoplasm and nucleus of Mtb-infected cells (Fig. 2a). We then investigated the underlying mechanism by which PtpA enters host cell nucleus. So far, the best-characterized nuclear import pathways are the importin α/β-dependent pathway and the Ras-related nuclear protein (Ran) guanosine diphosphate (RanGDP)/ankyrin repeat (RaDAR) pathway[29,30]. Since the localization of PtpA was not affected by ivermectin, a specific inhibitor of importin α/β-mediated nuclear import (Supplementary Fig. 3a), we thus excluded the possibility that PtpA enters host cell nucleus through the importin α/β-dependent pathway. Then we sought to determine whether RaDAR nuclear import pathway is involved in PtpA nuclear entry through an in vitro nuclear import assay, and our data showed that PtpA did not enter host cell nucleus through passive diffusion like free ubiquitin (Ub) but had a noticeable shift to the nucleus in the presence of the nuclear transport complex comprising RanGDP and nuclear transport factor 2 (NTF2) (Fig. 2b), indicating that PtpA enters host cell nucleus depending on the complex comprising RanGDP and NTF2 in the RaDAR nuclear import pathway.

We then sought to identify the specific sites of PtpA involved in its nuclear entry. As reported previously, proteins entering host cell nucleus through the RaDAR pathway usually contain the conserved ankyrin repeats (ARs) motif that can interact with RanGDP directly[29]. Through yeast two-hybrid (Y2H) interactome screening and immunoblot analysis, we found that PtpA directly interacted with Ran, its GDP-bound form (RanGDP), as well as its nuclear GTP-bound form (RanGTP), but only the interaction between PtpA and RanGDP was significantly enhanced upon treatment with NTF2 (Fig. 2c and Supplementary Fig. 3b), a nuclear import factor that directly interacts with RanGDP to induce a major conformational change in RanGDP to promote its interaction with AR-containing proteins[31]. To our surprise, we found that PtpA does not contain currently identifiable ARs motif according to an amino acid sequence analysis in the Simple Modular Architecture Research Tool (SMART) database, suggesting that PtpA might exploit alternative sites to interact with RanGDP for its nuclear

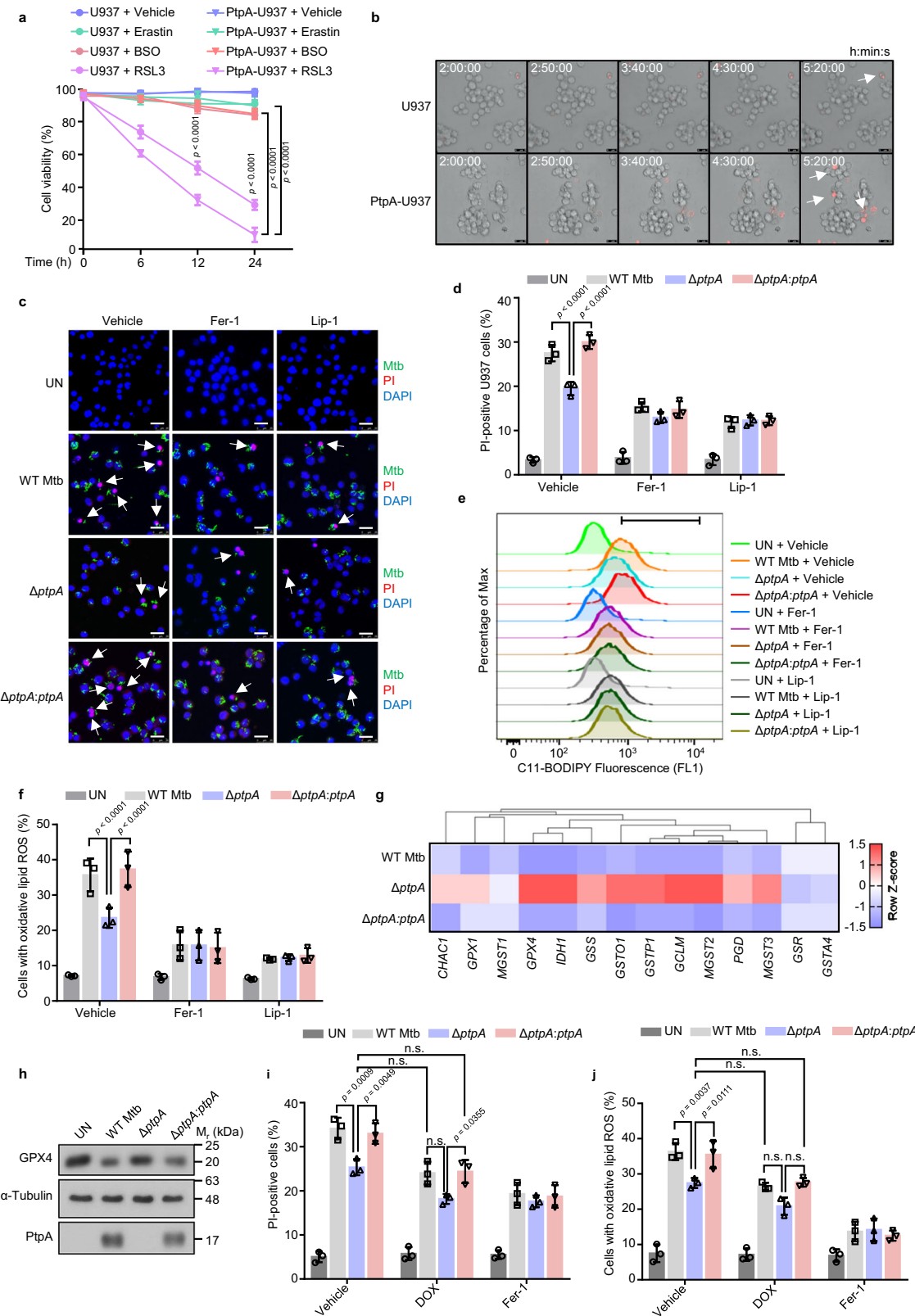

entry. It has been demonstrated that conserved residues are the key sites for binding to RanGDP[29], we then conducted multiple sequence alignment analysis for PtpA from several bacterial species, including Mtb, *Mycobacterium terrae*, *Mycobacterium marinum*, *Mycobacterium lepromatosis*, *Bifidobacterium bifidum*, and *Staphylococcus aureus*, and a total of four conserved residues including Cys11, Ile15, Leu62, and Phe135 were identified in PtpA (Supplementary Fig. 3c). By mutating

these residues or the phosphatase active site (Asp126) into Ala (C11A, I15A, L62A, D126A, and F135A), we found that only PtpA[C11A] mutant lost its ability to enter host cell nucleus (Fig. 2d, e). Meanwhile, the interaction between PtpA with RanGDP was abolished by the Cys11 mutation of PtpA (Fig. 2f). Given that Cys11 mutation affects both phosphatase activity[32] and nuclear import of PtpA, we then further examined whether the phosphatase activity and nuclear import ability

**Fig. 1 | Mtb PtpA induces GPX4-dependent ferroptosis. a** Cell viability of U937 cells by Cell Counting Kit-8 (CCK-8) assay. WT U937 cells and U937 cells stably expressing PtpA (PtpA-U937) were treated with vehicle, 10 μM Erastin, 50 μM BSO, or 2 μM RSL3 for the indicated time periods. **b** Frames of Supplementary Movies 1 and 2 showing propidium iodide (PI)-positive WT U937 and PtpA-U937 cells treated with 2 μM RSL3. Dead cells (red) were stained with PI. Scale bars, 25 μm. **c** Confocal microscopic analysis for cell death of U937 cells. Cells were uninfected (UN) or infected with WT Mtb, Mtb Δ*ptpA*, or Mtb Δ*ptpA:ptpA* strain at an MOI of 10 for 24 h with the treatment of vehicle, Fer-1, or Lip-1. Mtb strains (green) were stained with Alexa Fluor™ 488 succinimidyl ester, dead cells (red) were stained with PI, and nuclei (blue) were stained with DAPI. Scale bars, 25 μm. **d** Quantification of PI-positive infected U937 cells treated as in (**c**). **e** Flow cytometry for measurement of lipid peroxides with 1.5 μM BODIPY $C_{11}$ lipid probe in U937 cells treated as in (**c**). **f** Quantification of cells with oxidative lipid ROS treated as in (**c**). **g** Heat map of genes related to GSH metabolism in U937 cells infected with WT Mtb, Mtb Δ*ptpA*, or Mtb Δ*ptpA:ptpA* strain at an MOI of 10 for 24 h. The genes indicated in red and blue represent upregulated and down-regulated genes, respectively. **h** Immunoblot analysis of GPX4, α-Tubulin, and PtpA in U937 cells treated as in (**g**). **i** Quantification of PI-positive infected U937 cells. Cells were uninfected or infected with WT Mtb, Mtb Δ*ptpA*, or Mtb Δ*ptpA:ptpA* strain at an MOI of 10 for 24 h with the treatment of 1 μM DOX or 10 μM Fer-1. **j** Quantification of cells with oxidative lipid ROS treated as in (**i**). Error bars are means ± SD of three groups. Statistical significance was determined using two-way ANOVA (Tukey's multiple comparisons test). n.s., not significant. Source data are provided as a Source Data file.

of PtpA are related to ferroptosis induction. Through cell viability assay, we found that Mtb Δ*ptpA:ptpA*[CIIA] strain (which loses its phosphatase activity[32] and nuclear entry ability), but not Mtb Δ*ptpA:ptpA*[D126A] strain (which loses its phosphatase activity[32] but retains its nuclear entry ability), lost the activity to induce ferroptosis (Fig. 2g, h). Since there is a potential confounding issue of reduced bacterial load in cellular infection assays for ferroptosis induction, to further confirm that the functional deficiency of the PtpA mutant strain Δ*ptpA:ptpA*[CIIA], rather than the reduced bacteria load, is the cause of impaired ferroptosis, we further performed bacterial colony-forming units (CFUs) assay, and our data showed that the mutant strains including Mtb Δ*ptpA*, Mtb Δ*ptpA:ptpA*[D126A], and Mtb Δ*ptpA:ptpA*[CIIA] all had a significant decrease in Mtb survival in U937 cells as compared with that of WT Mtb strain and the complemented Mtb Δ*ptpA:ptpA* strain (Supplementary Fig. 3d). Together, these results indicate that PtpA enters host cell nucleus to trigger ferroptosis via its RanGDP-binding site Cys11, independent of its phosphatase activity and the bacterial load.

### The nuclear PtpA promotes ferroptosis via targeting arginine methyltransferase PRMT6

Our previous chromatin immunoprecipitation (ChIP)-sequencing data showed that the nuclear PtpA directly interacts with host DNA to regulate the transcription of genes mainly associated with cell proliferation, cell migration, innate immune responses, and intracellular membrane trafficking[27], we then re-analyzed this ChIP-sequencing data and found that PtpA did not target the promoter region of *GPX4*, suggesting that PtpA might have additional regulatory roles in host cell nucleus. To characterize the mechanism by which PtpA drives ferroptosis, several candidate PtpA-interacting proteins mainly located in host cell nucleus were isolated by the Y2H interactome screen (Supplementary Fig. 4a). Among them, the arginine methyltransferase PRMT6 has been reported to be involved in the regulation of many cellular processes including cell death[33,34], but whether PRMT6 is involved in the regulation of PtpA-mediated ferroptosis and the underlying molecular mechanism remains unexplored. Through the Y2H interactome screening experiment and immunoblot assay, we confirmed that PtpA directly interacted with PRMT6 (Fig. 3a and Supplementary Fig. 4b). Consistently, confocal microscopic analysis showed that PRMT6 was mainly located in host cell nucleus and co-localized with PtpA (Fig. 3b, c). To further determine whether PRMT6 is implicated in PtpA-induced ferroptosis, we constructed the *PRMT6*-knockout (*PRMT6*[−/−]) U937 cells by CRISPR/Cas9 system (Fig. 3d), and our results suggested that WT (*PRMT6*[+/+]) U937 cells were more sensitive to RSL3-induced ferroptosis and exhibited higher lactate dehydrogenase (LDH) release as well as the level of lipid peroxides, compared with *PRMT6*[−/−] U937 cells (Fig. 3e–h and Supplementary Fig. 4c), indicating that PRMT6 is involved in ferroptosis. During infection, U937 cells infected with WT Mtb or Mtb Δ*ptpA:ptpA* strain showed less resistance to ferroptosis accompanied by elevated LDH release and accumulation of lipid peroxides, compared with cells infected with Mtb Δ*ptpA* strain. Meanwhile, PtpA lost its ability to induce ferroptosis in *PRMT6*[−/−] U937 cells (Fig. 3i–k). Collectively, Mtb

PtpA promotes ferroptosis by targeting arginine methyltransferase PRMT6.

### Mtb PtpA inhibits GPX4 expression by promoting PRMT6-mediated H3R2me2a

H3R2me2a has been well recognized as the predominant histone mark catalyzed by methyltransferase PRMT6[35,36]. To examine how PtpA regulates the function of PRMT6, we detected the stability and methyltransferase activity of PRMT6 during Mtb infection, and we found that PtpA promoted asymmetric dimethylation of H3R2, without affecting the stability of PRMT6 (Fig. 4a). We then further examined the change of H3R2me2a levels in the ferroptotic process to verify whether PRMT6 methyltransferase activity is responsible for ferroptosis. As expected, PRMT6 expression and H3R2 methylation, increased in a dose- and time-dependent manner over the course of RSL3-induced ferroptosis (Supplementary Fig. 5a, b). Furthermore, although PRMT6[KLA], a methylase-inactive mutant[37], was capable of binding to PtpA like WT PRMT6 (Fig. 4b), PRMT6[KLA]-overexpressed cells exhibited higher cell viability and lower accumulation of lipid peroxides compared with that in *PRMT6*-overexpressed cells (Fig. 4c–f). Consistently, U937 cells treated with EPZ020411, an inhibitor of the methyltransferase activity of PRMT6[38], were resistant to ferroptosis with reduced lipid peroxides (Supplementary Fig. 5c–f). Meanwhile, PtpA lost its ability to induce ferroptosis in the presence of EPZ020411 during Mtb infection (Fig. 4g, h). These data imply that PRMT6 depends on its methyltransferase activity to mediate PtpA-induced ferroptosis.

To further explore how PtpA regulates ferroptosis by promoting PRMT6-mediated H3R2me2a, we examined the transcription level of genes in cysteine-GSH-GPX4 axis. Strikingly, PRMT6 specifically decreased the expression of *GPX4* depending on its methyltransferase activity (Supplementary Fig. 5g). During Mtb infection, WT Mtb or Mtb Δ*ptpA:ptpA* strain, but not Mtb Δ*ptpA* strain, reduced *GPX4* expression in *PRMT6*[+/+] U937 cells, which phenomenon was abolished in *PRMT6*[−/−] U937 cells (Fig. 4i, j). These results indicate that PtpA suppresses the expression of *GPX4* by targeting PRMT6. To investigate whether PRMT6-mediated H3R2me2a directly regulates *GPX4* transcription, we conducted ChIP analysis for H3R2me2a. As expected, H3R2me2a was enriched in the promoter of the *GPX4* gene. Moreover, the occupancy of H3R2me2a on the promoter of the *GPX4* gene was increased in *PRMT6*[+/+] U937 cells infected with WT Mtb or Mtb Δ*ptpA:ptpA* strain, which phenomenon disappeared in *PRMT6*[−/−] U937 cells (Fig. 4k), indicating that *GPX4* might represent an important downstream target gene of H3R2me2a. Together, our data demonstrate that PtpA promotes PRMT6-mediated H3R2me2a to inhibit the transcription of *GPX4*.

### Mtb PtpA enhances the methyltransferase activity of PRMT6

Given that PtpA promotes the asymmetric dimethylation of H3R2, we further explored the molecular mechanism by which PtpA regulates the methyltransferase activity of PRMT6. The exogenous addition of Mtb PtpA increased the asymmetric dimethylation of H3R2 in a dose-

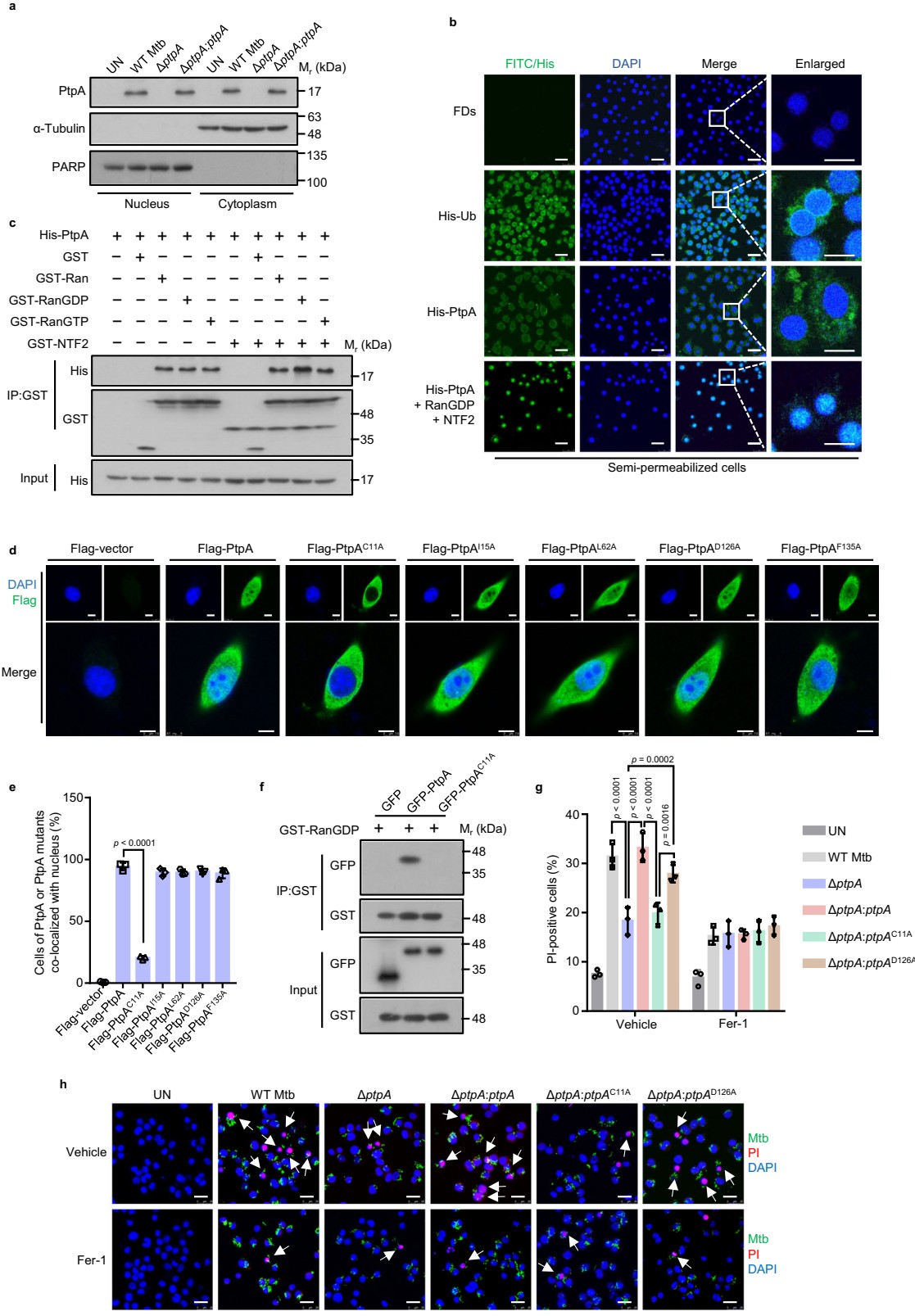

dependent manner without impacting the stability of PRMT6 (Fig. 5a). Moreover, Mtb PtpA could promote the methylation of PRMT6 on the substrate Benzoyl-L-arginine-7-amido-4-methylcoumarin (Bz-Arg-AMC), which activity was abolished in the absence of PRMT6 in vitro (Supplementary Fig. 6a), suggesting that PtpA could directly enhance the methyltransferase activity of PRMT6. In addition, we found that PtpA interacted with both PRMT6 and its physiological substrate

histone H3, thus enhancing the interaction of PRMT6 with histone H3 (Supplementary Fig. 6b). To further determine the specific regions by which PtpA interacts with PRMT6 or histone H3, we constructed PtpA truncated mutants including amino acid residues (1–50)-deleted protein (PtpA$^{\Delta 1-50}$), amino acid residues (51–70)-deleted protein (PtpA$^{\Delta 51-70}$), amino acid residues (70–122)-deleted protein (PtpA$^{\Delta 70-122}$), and amino acid residues (122–163)-deleted protein (PtpA$^{\Delta 122-163}$)

**Fig. 2 | Mtb PtpA, through a non-canonical RanGDP-binding site, enters host cell nucleus to induce ferroptosis. a** Immunoblot analysis of the cytosolic fraction and the pellets containing the nuclei in U937 cells. Cells were uninfected or infected with WT Mtb, Mtb $\Delta ptpA$, or Mtb $\Delta ptpA:ptpA$ strain at an MOI of 20 for 8 h. **b** Confocal microscopic analysis for protein nuclear localization in digitonin-treated semipermeable U937 cells. Cells were incubated with FITC-labeled dextrans (FDs, a negative control), His-Ub, or His-PtpA with or without GST-RanGDP and GST-NTF2 in the presence of 10 mg/mL U937 cytosol and an energy-regenerating mixture. His-tagged proteins (green) were stained with anti-His antibody, and nuclei (blue) were stained with DAPI. Scale bars, 25 μm. **c** Pull-down assay of purified His-PtpA by GST, GST-Ran, GST-RanGDP, GST-RanGTP with or without GST-NTF2. **d** Confocal microscopic analysis for nuclear localization of PtpA or PtpA mutants. A549 Cells were transfected with empty vector, or vectors encoding Flag-PtpA, Flag-PtpA$^{C11A}$, Flag-PtpA$^{I15A}$, Flag-PtpA$^{L62A}$, Flag-PtpA$^{D126A}$, or Flag-PtpA$^{F135A}$. Flag-

tagged proteins (green) were stained with anti-Flag antibody, and nuclei (blue) were stained with DAPI. Scale bars, 7.5 μm. **e** Quantification of cells of PtpA or PtpA mutants co-localized with host cell nuclei treated as in (**d**). **f** Pull-down assay of GFP, GFP-PtpA, or GFP-PtpA$^{C11A}$ by GST-RanGDP. **g** Quantification of PI-positive infected U937 cells. Cells were incubated with 5 μg/mL PI for 20 min after being infected with WT Mtb, Mtb $\Delta ptpA$, Mtb $\Delta ptpA:ptpA$, Mtb $\Delta ptpA:ptpA^{C11A}$, or Mtb $\Delta ptpA:pt-pA^{D126A}$ strain at an MOI of 10 for 24 h with the treatment of vehicle or Fer-1. Uninfected cells were used as the control. **h** Confocal microscopic analysis for cell death of U937 cells treated as in (**g**). Mtb strains (green) were stained with Alexa Fluor™ 488 succinimidyl ester, dead cells (red) were stained with PI, and nuclei (blue) were stained with DAPI. Scale bars, 25 μm. Error bars are means ± SD of three groups. Statistical significance was determined using one-way ANOVA (Dunnett's multiple comparisons test) and two-way ANOVA (Tukey's multiple comparisons test). Source data are provided as a Source Data file.

(Fig. 5b). Our data showed that the N-terminal region (amino acid: 1–50) of PtpA was responsible for its interaction with PRMT6, and the C-terminal region (amino acid: 122–163) interacted with histone H3 (Fig. 5c). Consistently, both PtpA$^{\Delta1-50}$ and PtpA$^{\Delta122-163}$ decreased the level of H3R2me2a during Mtb infection (Fig. 5d). Together, PtpA not only directly enhances the methyltransferase activity of PRMT6, but also increases the affinity between PRMT6 and histone H3, thereby leading to enhanced asymmetric dimethylation of H3R2.

To confirm whether PtpA induces ferroptosis through interacting with PRMT6, we then detected the viability of cells overexpressing WT PtpA, PtpA$^{\Delta1-50}$, or PtpA$^{\Delta122-163}$. We found that PtpA$^{\Delta1-50}$ largely abrogated, while PtpA$^{\Delta122-163}$ partially abrogated, the activity to induce ferroptosis, as compared with WT PtpA (Fig. 5e, f), suggesting that PtpA-induced ferroptosis is mainly dependent on its PRMT6-interacting N-terminal region. Interestingly, we also noticed that the PtpA$^{\Delta1-50}$ mutant could enter host cell nucleus independent of its C11 site. To explore the mechanism by which the PtpA$^{\Delta1-50}$ mutant enters the nucleus of host cells, we conducted an in vitro nuclear import assay, and found that PtpA$^{\Delta1-50}$ had a noticeable shift to the nucleus in the absence of RanGDP and NTF2 (Supplementary Fig. 6c), suggesting that PtpA$^{\Delta1-50}$, with a lower molecular weight of 12 KDa, can freely diffuse into the host cell nucleus, independent of the RanGDP/NTF2 complex-mediated nuclear import system. We then constructed the Mtb $\Delta ptpA:ptpA^{\Delta1-50}$ mutant strain for infection of U937 cells and confirmed that the PtpA$^{\Delta1-50}$ protein was located in both cytoplasm and nucleus of host cells (Supplementary Fig. 6d, e). We then demonstrated that the PtpA$^{\Delta1-50}$ mutant lost its phosphatase activity, thus failing to inhibit phagosome acidification and innate immune signaling pathway activation[28,32] (Supplementary Fig. 6f–h). Furthermore, WT Mtb and Mtb $\Delta ptpA:ptpA$ strains, but not Mtb $\Delta ptpA$ and Mtb $\Delta ptpA:ptpA^{\Delta1-50}$ strains, suppressed GPX4 expression and promoted ferroptosis, LDH release, as well as lipid peroxides in PRMT6$^{+/+}$ U937 cells, which effects were abolished in PRMT6$^{-/-}$ U937 cells (Fig. 5g–j). Altogether, these results indicate that Mtb PtpA interacts with PRMT6 to enhance its methyltransferase activity, thus promoting host ferroptosis.

**Mtb PtpA induces ferroptosis to promote pathogen pathogenicity and dissemination in vivo**

Ferroptosis usually drives tissue necrosis to help Mtb pathogenicity and dissemination in mice[8,9]. As reported previously, iron homeostasis contributing to ferroptosis is distinct between different mouse strains. At basal levels, BALB/c mice show significantly higher levels of iron compared with C57BL/6 mice[39], we thus assessed the impact of PtpA-induced ferroptosis on pulmonary necrosis and Mtb dissemination using the BALB/c mice treated with ferroptosis inhibitor Lip-1. As expected, mice infected with WT Mtb or Mtb $\Delta ptpA:ptpA$ strain showed more severe pathological lung damage and higher abundance of acid-fast bacilli in the lungs (Fig. 6a, b and Supplementary Fig. 7a), as well as higher load of mycobacteria in lungs and spleens than mice infected with Mtb $\Delta ptpA$ or Mtb $\Delta ptpA:ptpA^{\Delta1-50}$ strain (Fig. 6c, d). Consistent

with the in vitro data, mice infected with WT Mtb or Mtb $\Delta ptpA:ptpA$ strain exhibited increased H3R2me2a level (Fig. 6e, f), decreased GPX4 expression, and elevated 4-hydroxy-2-noneal (4-HNE, a ferroptosis marker) expression, compared with that in mice infected with Mtb $\Delta ptpA$ or Mtb $\Delta ptpA:ptpA^{\Delta1-50}$ strain (Fig. 6g, h and Supplementary Fig. 7b, c). Meanwhile, ferroptosis inhibitors could significantly diminish the above differences between WT Mtb (or Mtb $\Delta ptpA:ptpA$)-infected and Mtb $\Delta ptpA$ (or Mtb $\Delta ptpA:ptpA^{\Delta1-50}$)-infected mice. Collectively, Mtb PtpA aggravates ferroptosis-induced lung injury and Mtb dissemination depending on its PRMT6-binding region.

In summary, our study reveals the detailed molecular mechanism by which PtpA enters host cell nucleus to induce host ferroptosis. Specifically, PtpA enters the nucleus of host cells depending on a conserved Cys site, which is responsible for interacting with RanGDP. The nuclear PtpA then directly interacts with PRMT6 to increase asymmetric dimethylation of H3R2, thereby inhibiting the expression of GPX4, eventually inducing ferroptosis-induced Mtb pathogenicity and dissemination (Fig. 6i and Supplementary Fig. 8).

## Discussion

Pathogenic organisms manipulate a variety of mechanisms to evade and co-opt immune responses to achieve their intracellular survival and dissemination[40,41]. To date, the majority of studies have focused on the mechanisms by which pathogens infect the host cell and survive within it, but how pathogens achieve their dissemination is largely unknown. The programmed necrosis has been revealed to be a critical contributor to promote tissue damage and pathogen dissemination. Ferroptosis is a newly defined iron-dependent programmed necrosis induced by lipid peroxidation, and the previous studies mainly focus on the role of ferroptosis in cancer development and provide perspective targets for cancer therapy[42,43]. Recently, a few studies have suggested that ferroptosis participates in regulating infection-induced tissue damage and bacterial dissemination[7,8]. However, the specific pathogen effectors and the underlying molecular mechanisms that regulate ferroptosis to promote tissue damage and pathogen dissemination remain poorly understood. TB, a life-threatening chronic infectious disease caused by Mtb, is characterized by the formation of caseous necrosis[44], which has been recently linked to Mtb pathogenicity and dissemination without being fully understood. Here, we identify Mtb PtpA as a strong pro-ferroptotic effector in Mtb-induced ferroptosis. Previously, PtpA is reported to be a critical effector in promoting cell proliferation and inhibiting phagosome acidification as well as immune signaling pathway, depending on its tyrosine phosphatase and DNA-binding activities[27,28,45,46]. Here, we reveal an additional function of PtpA in triggering ferroptosis by mediating the epigenetic modification in host cell nucleus, which is independent of its tyrosine phosphatase and DNA-binding activities. Specifically, the nuclear PtpA interacts with PRMT6 to enhance its methyltransferase activity and PRMT6-mediated H3R2me2a, thus inhibiting GPX4 expression, followed by the induction of ferroptosis.

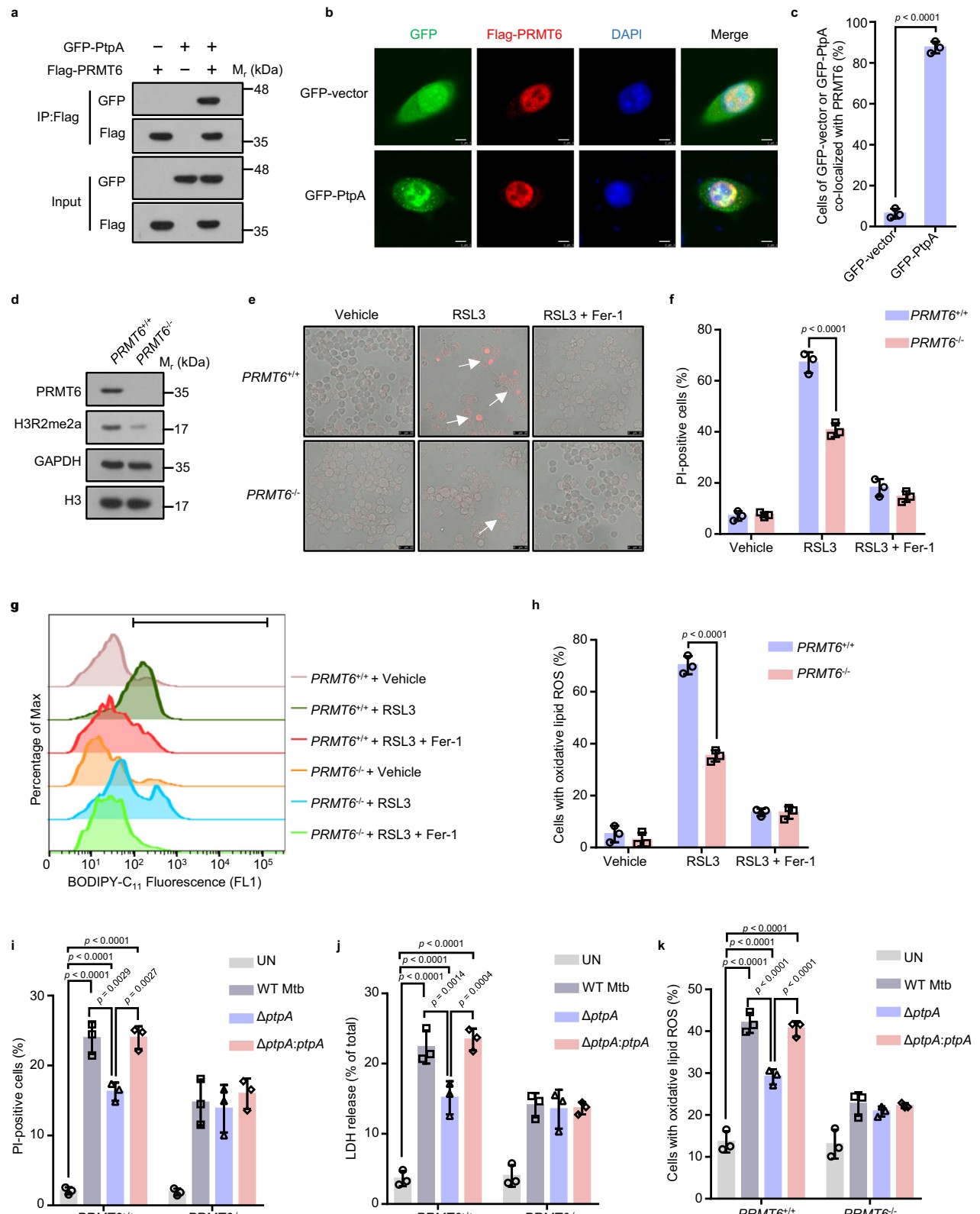

Multiple pathogens have been found to disturb various host cellular processes upon host nuclear entry[47,48]. Most bacterial effectors harbor eukaryotic nuclear localization signals (NLS) and cross the host nuclear envelope barrier depending on host importin α/β[49,50]. Here we found that Mtb PtpA enters host cell nucleus via interacting with host RanGDP, but not importin α/β. Also worth mentioning is that proteins entering the nucleus by the RaDAR pathway usually directly interact with RanGDP via a conserved ARs motif[29]. Interestingly, PtpA binds to RanGDP depending on its Cys11 site instead of a conserved ARs motif. Furthermore, the Cys11 site in Mtb PtpA is highly conserved among pathogenic bacteria, such as *Mycobacterium lepromatosis* and *Staphylococcus aureus*, suggesting that the non-canonical RanGDP-binding Cys site may serve as a potential target against pathogenic bacteria by preventing pathogen effectors from entering host cell nucleus.

**Fig. 3 | Nuclear Mtb PtpA promotes ferroptosis via targeting PRMT6. a** Co-immunoprecipitation of GFP-PtpA from the lysates of HEK293T cells co-transfected with Flag-PRMT6. **b** Confocal microscopic analysis for the subcellular localization of GFP or GFP-PtpA (green) with Flag-PRMT6 in A549 cells. Flag-tagged proteins (red) were stained with anti-Flag antibody, and nuclei (blue) were stained with DAPI. Scale bars, 8 μm. **c** Percentage of cells of GFP or GFP-PtpA co-localized with PRMT6 treated as in (**b**). **d** Immunoblot analysis of PRMT6, H3R2me2a, GAPDH, and histone H3 in $PRMT6^{+/+}$ and $PRMT6^{-/-}$ U937 cells. **e** Representative phase-contrast images of PI-positive $PRMT6^{+/+}$ or $PRMT6^{-/-}$ U937 cells treated with vehicle, 2 μM RSL3, or 10 μM Fer-1 for 8 h. Dead cells (red) were stained with PI. Scale bars, 25 μm. **f** Quantification of PI-positive $PRMT6^{+/+}$ or $PRMT6^{-/-}$ U937 cells treated as in (**e**). **g** Flow cytometry analysis for lipid peroxides using 1.5 μM BODIPY $C_{11}$ lipid probe in U937 cells. Cells were treated with vehicle, 2 μM RSL3, or 10 μM Fer-1 for 8 h. **h** The percentage of cells with oxidative lipid ROS treated as in (**g**). **i** Quantification of PI-positive $PRMT6^{+/+}$ or $PRMT6^{-/-}$ U937 cells uninfected or infected with WT Mtb, Mtb $\Delta ptpA$, or Mtb $\Delta ptpA:ptpA$ strain at an MOI of 10 for 24 h. **j** LDH release of $PRMT6^{+/+}$ or $PRMT6^{-/-}$ U937 cells treated as in (**i**). **k** The percentage of $PRMT6^{+/+}$ or $PRMT6^{-/-}$ U937 cells with oxidative lipid ROS. Cells were uninfected or infected with WT Mtb, Mtb $\Delta ptpA$, or Mtb $\Delta ptpA:ptpA$ strain at an MOI of 10 for 24 h, and incubated with 1.5 μM BODIPY $C_{11}$ lipid probe for 20 min. Error bars are means ± SD of three groups. Statistical significance was determined using unpaired two-sided Student's *t* test, one-way ANOVA (Dunnett's multiple comparisons test) and two-way ANOVA (Tukey's multiple comparisons test). Source data are provided as a Source Data file.

Epigenetic modifications are pivotal for regulating gene expression in response to extracellular stimuli. During infection, intracellular pathogenic bacteria play a profound effect on the host epigenome to promote their intracellular survival and cause diseases[51,52]. Mtb, as an extremely successful intracellular pathogen, can evade host immune responses for intracellular survival by adopting a plethora of strategies, one such strategy is to alter the host epigenome to modulate the transcriptional machinery[53]. For example, a few studies reveal that Mtb effector Rv1988 and Rv3423.1 methylate H3R42 and H3K9/K14 respectively, thus helping Mtb escape from host immune defense and survive within host cells[26,54]. Except for ensuing its intracellular survival, Mtb also facilitates its dissemination and causes severe pathology under certain circumstances. However, the specific molecular mechanisms by which and how Mtb effectors target host epigenetic modification to cause immunopathologic injury and facilitate its dissemination remain largely unknown. Here, we demonstrate that Mtb PtpA directly interacts with host PRMT6 to enhance its methyltransferase activity, leading to increased asymmetric dimethylation of H3R2 to inhibit GPX4 expression, thereby promoting the occurrence of ferroptosis and Mtb dissemination. It is worth mentioning that ferroptosis is usually regulated by multiple signaling pathways including cysteine-GSH-GPX4 axis, iron metabolism, reactive oxygen species (ROS) metabolism, and MAPK pathway, among which the cysteine-GSH-GPX4 axis is identified as the classical core regulator of ferroptosis[3,4]. Here, we reveal that PtpA triggers ferroptosis by directly disrupting the core axis of ferroptosis via targeting PRMT6, indicating that PtpA is a critical pro-ferroptotic effector in host-pathogen interaction. It should also be mentioned that we noticed that inhibition of lipid peroxidation (through treatment with Fer-1) was more effective than GPX4 restoration in suppressing Mtb-induced ferroptosis. Since multiple signaling pathways, including cysteine-GSH-GPX4 axis, iron metabolism, ROS metabolism, and MAPK pathway, could promote the production of lipid peroxides to induce ferroptosis[3,4], we thus speculated that besides Mtb PtpA-GPX4 axis-mediated ferroptosis activation, Mtb might regulate host ferroptosis partially by targeting additional pathways, such as iron metabolism, reactive oxygen species (ROS) metabolism, and MAPK pathway. Collectively, our findings provide unrecognized insights into molecular mechanisms of pathogen-induced ferroptosis, indicating a potential TB treatment via blocking Mtb PtpA-host PRMT6 interface to target GPX4-dependent ferroptosis.

## Methods
### Ethics
All animal studies were approved by the Biomedical Research Ethics Committee of Institute of Microbiology, Chinese Academy of Sciences (SQIMCAS2018005). PBMCs used in this study were collected from healthy volunteers, and the study was approved by the Ethics Committee of Beijing Chest Hospital, Capital Medical University. Informed consent was obtained from all individual participants included in the study.

### Bacterial strains
*Mycobacterium smegmatis* mc²155 (ATCC, 700084), *Mycobacterium bovis* BCG (ATCC, 35734), Mtb H37Rv (ATCC, 27294), Mtb $\Delta ptpA$, Mtb $\Delta ptpA:ptpA$, Mtb $\Delta ptpA:ptpA^{C11A}$, Mtb $\Delta ptpA:ptpA^{D126A}$, Mtb $\Delta ptpA:ptpA^{\Delta 1-50}$, and Mtb $\Delta ptpA:ptpA^{\Delta 122-163}$ were grown in Middlebrook 7H9 medium (BD, 271310) supplemented with 10% oleic acid-albumin-dexrose-catalase (OADC) and 0.05% Tween-80 (G-CLONE, CS9029), or on Middlebrook 7H10 agar (BD, 262710) supplemented with 10% OADC.

### Cell culture
The A549 (ATCC, CCL-185) and HEK293T cells (ATCC, CRL-3216) were cultured in Dulbecco's modified Eagle's medium (DMEM, Gibco, C11995500BT-1) supplemented with 10% fetal bovine serum (FBS, Gibco, 10091148). The U937 cells (ATCC, CRL-1593.2) were maintained in RPMI-1640 (Gibco, C11875500BT) with 10% FBS and differentiated into adherent macrophage-like cells with 10 ng/mL phorbol 12-myristate 13-acetate (PMA) overnight, then the cells were washed once with PBS and cultured in fresh RPMI-1640 medium.

### Reagents and antibodies
The following chemicals were used: BODIPY (581/591) $C_{11}$ (Cayman, 27086), PMA (Sigma-Aldrich, P1585), DAPI (Beyotime, C1002), PI (Beyotime, ST511), Erastin (Selleck, S7242), BSO (Selleck, S9728), RSL3 (Selleck, S8155), Fer-1 (Selleck, S7243), Lip-1 (MCE, HY-12726), EPZ020411 2HCl (Selleck, S7820), ATP (Roche, 11140965001), GTP (Macklin, G6195), Digitonin (Sigma-Aldrich, D5628), S-(5′-Adenosyl)-L-methionine iodide (Sigma-Aldrich, A4377), Creatine phosphate (Roche, 10621714001), Creatine kinase (Roche, 10736988001), Lipotectamine 2000 (Invitrogen, 11668019), Bz-Arg-AMC (Bachem, 4002540.0050), FITC-labeled dextrans (Sigma-Aldrich, 90718), tetracycline (Amresco, 0422), Alexa Fluor 488 succinimidyl ester (Invitrogen, A20000), pHrodo Red succinimidyl ester (Invitrogen, P36600). For the antibodies, Anti-Flag antibody (1:200 for immunofluorescence, Sigma-Aldrich, F1804), Anti-Ag85 antibody (1:1000 for immunoblotting, Abcam, ab36731), Anti-Mtb antibody (1:200 for immunofluorescence, Abcam, ab905), Anti-DDDDK-tag-pAb-HRP-DirecT (1:2000 for immunoblotting, MBL, PM020-7), Anti-GFP-tag-pAb-HRP-DirecT (1:1000 for immunoblotting, MBL, 598-7), Anti-H3R2me2a antibody (1:1000 for immunoblotting, Abcam, ab194706), Anti-H3R2me2a antibody (1:200 for immunofluorescence, ABclonal, A3155), Anti-PRMT6 antibody (1:1000 for immunoblotting, Cell Signaling Technologies, 14641S), Anti-H3 antibody (1:1000 for immunoblotting, Abcam, ab1791), Anti-α-Tubulin antibody (1:2000 for immunoblotting, Sigma-Aldrich, T6199), Anti-GPX4 antibody (1:1000 for immunoblotting and 1:200 for immunohistochemistry, ABclonal, A11243), Anti-GST antibody (1:1000 for immunoblotting, ABclonal, AE006), Anti-PARP antibody (1:1000 for immunoblotting, Cell Signaling Technologies, 9542), Anti-His antibody (1:1000 for immunoblotting, ABclonal, AE003), Anti-4 Hydroxynonenal antibody (1:200 for immunohistochemistry, Abcam, ab46545), Anti-GAPDH antibody (1:5000 for immunoblotting, Santa Cruz, sc-25778), Anti-JNK antibody (1:1000 for immunoblotting, Cell Signaling

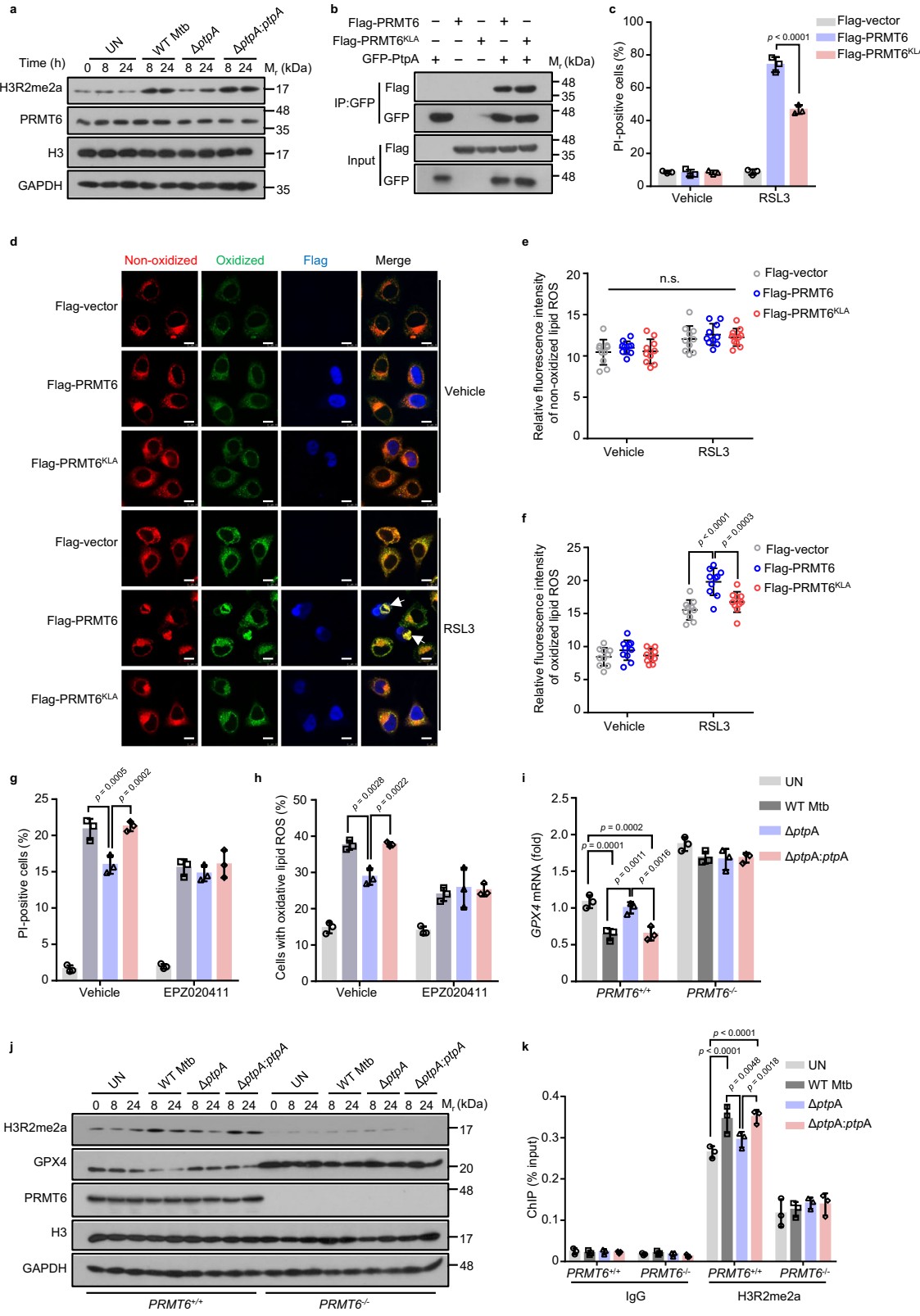

Technologies, 9252), Anti-p-JNK antibody (1:1000 for immunoblotting, Cell Signaling Technologies, 9255), Anti-p38 antibody (1:1000 for immunoblotting, Cell Signaling Technologies, 9212), Anti-p-p38 antibody (1:1000 for immunoblotting, Cell Signaling Technologies 9211), Goat-anti-Mouse HRP (1:10000 for immunoblotting, ZSGB-BIO, ZB-2305), Goat-anti-Rabbit HRP (1:10000 for immunoblotting, ZSGB-BIO, ZB-2306), Goat-anti-Mouse FITC (1:200 for immunofluorescence, ZSGB-BIO, ZF-0312), Goat-anti-Mouse Alexa Fluor 594 (1:200 for immunofluorescence, ZSGB-BIO, ZF-0513), Goat anti-Mouse IgG (H + L) Secondary Antibody, DyLight 405 (1:200 for immunofluorescence, Invitrogen, 35501BID), Anti-PtpA antibody was produced and purified by GenScript Biotechnology with the recombinant GST-tagged PtpA protein as the immunogen (1:1000 for immunoblotting and 1:200 for immunofluorescence)[28].

**Fig. 4 | Mtb PtpA inhibits GPX4 expression by promoting PRMT6-mediated H3R2me2a. a** Immunoblot analysis of H3R2me2a, PRMT6, H3, and GAPDH in U937 cells. Cells were uninfected or infected with WT Mtb, Mtb ΔptpA, or Mtb ΔptpA:ptpA strain at an MOI of 10 for 24 h. **b** Co-immunoprecipitation of Flag-PRMT6 or Flag-PRMT6KLA from the lysates of HEK293T cells co-transfected with GFP-PtpA. **c** Quantification of PI-positive A549 cells. Cells transfected with empty vector, or vectors encoding GFP-PRMT6 or GFP-PRMT6KLA were treated with vehicle or 4 μM RSL3 for 8 h. **d** Confocal microscopic analysis for lipid peroxides using BODIPY C11 lipid probe in A549 cells. Cells transfected with empty vector, or vectors encoding Flag-PRMT6 or Flag-PRMT6KLA were treated with vehicle or 4 μM RSL3 for 8 h, and were then treated with 1.5 μM BODIPY C11 lipid probe for additional 20 min. Flag-tagged proteins (blue) were stained with anti-Flag antibodies. Red and green fluorescence represent the non-oxidized and oxidized BODIPY C11, respectively.

Scale bars, 10 μm. **e, f** Quantification of the fluorescence intensity of non-oxidized (**e**) and oxidized BODIPY C11 (**f**) in cells treated as in (**d**). **g** Quantification of PI-positive infected U937 cells. Cells were uninfected or infected with WT Mtb, Mtb ΔptpA, or Mtb ΔptpA:ptpA strain as in **a** after being treated with vehicle or 20 μM EPZ020411. **h** Quantification of cells with oxidative lipid ROS treated as in (**g**). **i** RT-qPCR analysis for the mRNA of GPX4 in PRMT6+/+ or PRMT6−/− U937 cells uninfected or infected with WT Mtb, Mtb ΔptpA, or Mtb ΔptpA:ptpA strain as in (**a**). **j** Immunoblot analysis of H3R2me2a, GPX4, PRMT6, H3, and GAPDH in PRMT6+/+ or PRMT6−/− U937 cells treated as in (**i**). **k** ChIP-qPCR analysis of the GPX4 promoter in PRMT6+/+ or PRMT6−/− U937 cells using the H3R2me2a antibody. Error bars are means ± SD of three groups. Statistical significance was determined using two-way ANOVA (Tukey's multiple comparisons test). n.s., not significant. Source data are provided as a Source Data file.

## Generation of Mtb mutants

The Mtb ΔptpA strain was constructed using specialized transduction as described previously[55]. This allelic exchange substrate designed to replace the ptpA gene was introduced into the PacI site of phasmid phAE159 and electroporated into *Mycobacterium smegmatis* mc²155 to obtain high-titers phage lysates. Mtb strain was then washed twice with MP buffer containing 50 mM Tris (pH 8.0), 150 mM NaCl, 10 mM MgCl₂, 2 mM CaCl₂, and mixed with phage lysates at 37 °C overnight. The cell resuspension was plated on Middlebrook 7H10 agar containing 75 μg/mL hygromycin at 37 °C. Immunoblot assay was used to confirm the deletion of ptpA in the Mtb strain. For ptpA and ptpA mutants complementation, the integrative vector pMV306[56] was used to create ΔptpA:ptpA, Mtb ΔptpA:ptpACllA, Mtb ΔptpA:ptpAD126A, ΔptpA:ptpAΔ1−50, and ΔptpA:ptpAΔ122−163 strains. The primers used were listed in Supplementary Data 1.

## Bacterial infection and CFU assay

The differentiated U937 cells were infected with Mtb strains at an MOI of 10. After 2 h, the infected cells were washed three times with pre-warmed PBS and incubated again with the fresh RPMI 1640 medium. For colony-forming unit (CFU) assay, the macrophages were lysed in 7H9 broth containing 0.05% SDS, and the lysates were plated on 7H10 agar at various time points. Colonies were then counted after 3–4 weeks.

## Mouse model

Male Specified Pathogen Free (SPF) BALB/c mice (approximately 6–8 weeks) were purchased from Vital River and maintained under barrier conditions in a BSL-3 biohazard animal room. All the mice were housed under SPF conditions (12 h light/dark cycle, 50% relative humidity, between 25 and 27 °C) with free access to food and tap water. Mice were challenged by aerosol exposure with Mtb H37Rv using an inhalation device (Glas-Col) calibrated to deliver 100 CFUs of Mtb. At day 15 post-infection, vehicle or Lip-1 (3 mg/kg) was injected intraperitoneally into mice daily. After 5 weeks of infection, lungs and spleens were harvested and sonicated in 1 mL PBS. About 100 μL of each sample was diluted and plated on 7H10 agar plates. Colonies were counted after 3-4 weeks of incubation at 37 °C. Some samples of lungs were fixed in 10% formalin and embedded in paraffin, and then sections were cut for hematoxylin and eosin staining, Ziehl-Neelsen acid-fast staining, immunofluorescence, or immunohistochemistry in blinded fashion by a pathologist. The tissue slices were collected by Leica CS2 and analyzed by SlideViewer (v2.5.0.143918), ImageScope (v12.3.3.7014) and ImageJ (v1.8.0) software. The sample size was based on empirical data from pilot experiments. No additional randomization or blinding was used to allocate experimental groups.

## Yeast two-hybrid assay

The Matchmaker Two-Hybrid System and Mtb ptpA gene cloned into the bait Gal4-BD vector pGBKT7 were used for the yeast two-hybrid assay as described[28]. To test the interaction between PtpA with PRMT6 and Ran separately, the pGBKT7-PtpA, pGADT7-PRMT6, and pGADT7-

Ran plasmids were separately transformed or co-transformed into Y2HGold competent yeast cells. The transformants were plated onto low-stringency (lacking leucine and tryptophan) and high-stringency (lacking adenine, histidine, leucine, and tryptophan) selection plates for detection.

## Knockout cell lines

The CRISPR-Cas9 gRNA design tool (https://crisprgold.mdc-berlin.de/) was selected to design the specific target sequences of human PRMT6 for sgRNA synthesis. The pairs of annealed oligos were cloned into the lentiCRISPRv2 vector after being digested with BsmBI (ThermoFisher, ER0451). After transfecting the cloned lentiviral vector, psPAX2, and pMD2.G into HEK293T cells for 48 h, the supernatants were collected to incubate with U937 cells. Single GFP-positive cell was sorted in 96-well plates by flow cytometry using the BD FACSAria III cell sorter (BD Biosciences), the single-cell-derived clones were expanded and verified by immunoblot analysis.

## Cell fractionation

Differentiated U937 cells were infected with Mtb strains at an MOI of 20 for 8 h. Cytosolic and nuclear proteins were extracted using a Subcellular Protein Fractionation Kit (ThermoFisher, 78833) according to the manufacturer's instructions and analyzed by immunoblot analysis.

## Immunoprecipitation and immunoblot analysis

Protein interactions were assessed using immunoprecipitation. Briefly, cells were washed once with cold PBS and lysed in cold NP-40 Lysis Buffer (Beyotime, P0013F) containing protease inhibitors. The supernatants of the cell lysates were isolated at 12,500 × g for 30 min at 4 °C and were then incubated with anti-Flag M2 Affinity Gel (Sigma-Aldrich, A2220) and GFP-Nanoab-Agarose (LABLEAD, GNA-25-500), followed by extensive washing with NP-40 Lysis Buffer. For immunoblot analysis, samples were run on SDS-PAGE gels. Proteins were separated by SDS-PAGE and transferred to polyvinylidene difluoride membranes (Millipore, IPVH00010). After incubation with primary and secondary antibodies, protein bands were detected using an Immobilon Western Chemiluminescent HRP Substrate (Millipore, WBKLS0500) and exposed to X-ray film.

## Semi-in vivo and in vivo precipitation assay

Fusion proteins including GST-PtpA, GST-PRMT6, GST-Ran, GST-RanT24N, GST-RanQ69L, GST-NTF2, and His-PtpA were purified from *Escherichia coli* as described[28]. For semi-in vivo precipitation assay, U937 cells were lysed in the NP-40 Lysis Buffer and were then incubated with H3 antibody overnight. Immunoprecipitation was performed with Protein A/G PLUS-Agarose (Santa cruz, sc-2003) for 3 h. The immunoprecipitated proteins were then incubated with purified GST-PtpA or GST-PRMT6. After being washed four times, the pellets were eluted with 60 μL of SDS loading buffer. For in vivo precipitation assays, 2–10 μg of proteins were immobilized onto glutathione

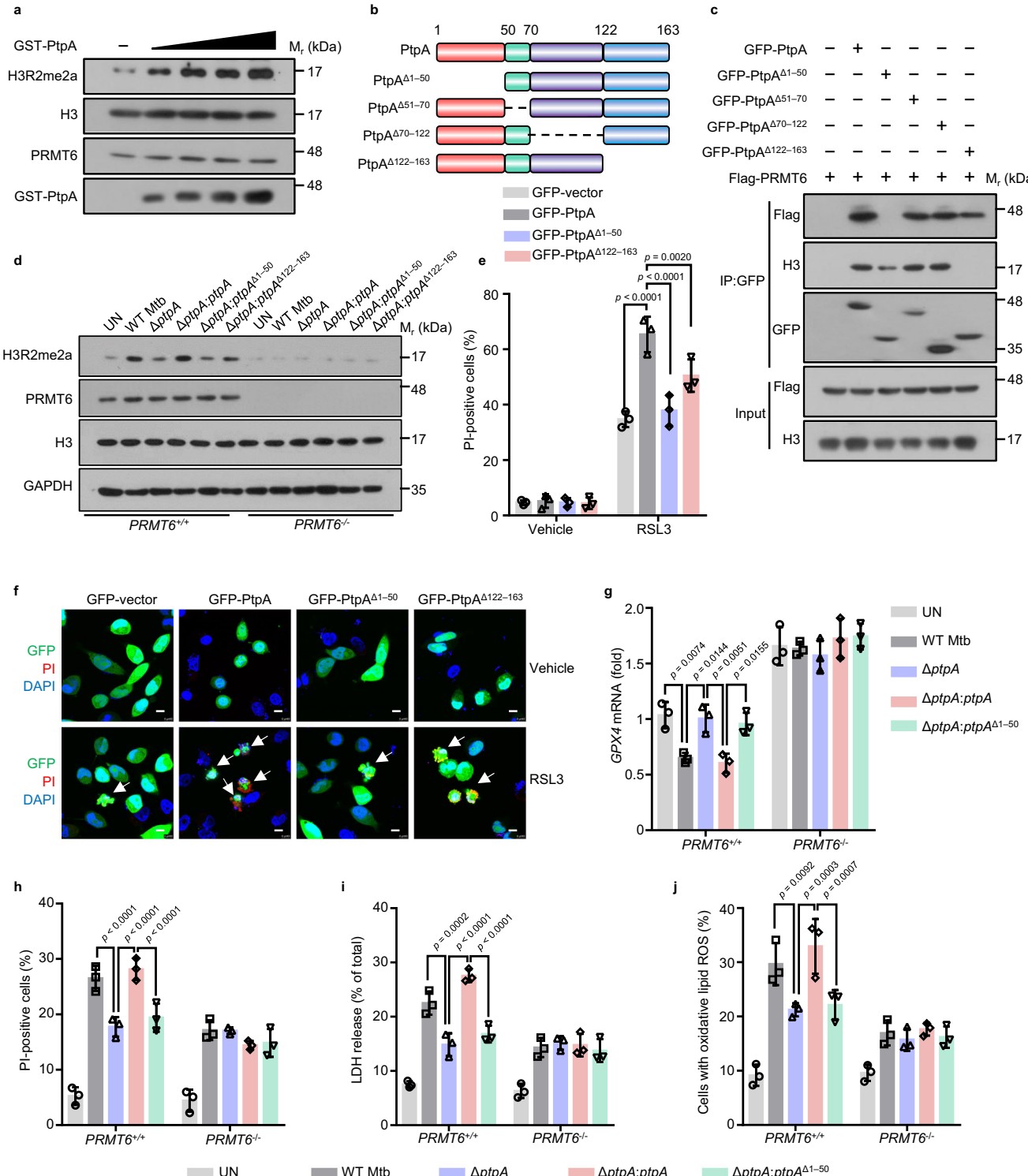

**Fig. 5 | Mtb PtpA enhances methyltransferase activity of PRMT6. a** Immunoblot analysis of H3R2me2a, H3, PRMT6, and GST-PtpA in HEK293T cells. Cell lysates were incubated with 0.1, 0.2, 0.4, or 0.8 μM GST-PtpA for 2 h. **b** Schematic representation of PtpA truncated mutants including PtpA, PtpA$^{\Delta 1-50}$, PtpA$^{\Delta 51-70}$, PtpA$^{\Delta 70-122}$, and PtpA$^{\Delta 122-163}$. **c** Co-immunoprecipitation of Flag-PRMT6 and H3 from the lysates of HEK293T cells co-transfected with GFP-PtpA, GFP-PtpA$^{\Delta 1-50}$, GFP-PtpA$^{\Delta 51-70}$, GFP-PtpA$^{\Delta 70-122}$, or GFP-PtpA$^{\Delta 122-163}$. **d** Immunoblot analysis of H3R2me2a, PRMT6, H3, and GAPDH in $PRMT6^{+/+}$ or $PRMT6^{-/-}$ U937 cells. Cells were uninfected or infected with WT Mtb, Mtb $\Delta ptpA$, Mtb $\Delta ptpA$:$ptpA$, Mtb $\Delta ptpA$:$ptpA^{\Delta 1-50}$, or Mtb $\Delta ptpA$:$ptpA^{\Delta 122-163}$ strain at an MOI of 10 for 24 h. **e** Quantification of PI-positive cells. A549 cells were transfected with GFP, GFP-PtpA, GFP-PtpA$^{\Delta 1-50}$, or GFP-PtpA$^{\Delta 122-163}$

for 24 h, and were then treated with vehicle or 2 μM RSL3 for 8 h. **f** Confocal microscopic analysis for cell death in A549 cells treated as in (**e**). **g** RT-qPCR analysis of *GPX4* mRNA in $PRMT6^{+/+}$ or $PRMT6^{-/-}$ U937 cells. Cells were uninfected or infected with WT Mtb, Mtb $\Delta ptpA$, Mtb $\Delta ptpA$:$ptpA$, or Mtb $\Delta ptpA$:$ptpA^{\Delta 1-50}$ strain at an MOI of 10 for 24 h. **h** Quantification of PI-positive $PRMT6^{+/+}$ or $PRMT6^{-/-}$ U937 cells treated as in (**g**). **i** LDH release of $PRMT6^{+/+}$ or $PRMT6^{-/-}$ U937 cells treated as in (**g**). **j** Quantification of $PRMT6^{+/+}$ or $PRMT6^{-/-}$ U937 cells with oxidative lipid ROS treated as in (**g**). Error bars are means ± SD of three groups. Statistical significance was determined using two-way ANOVA (Tukey's multiple comparisons test). Source data are provided as a Source Data file.

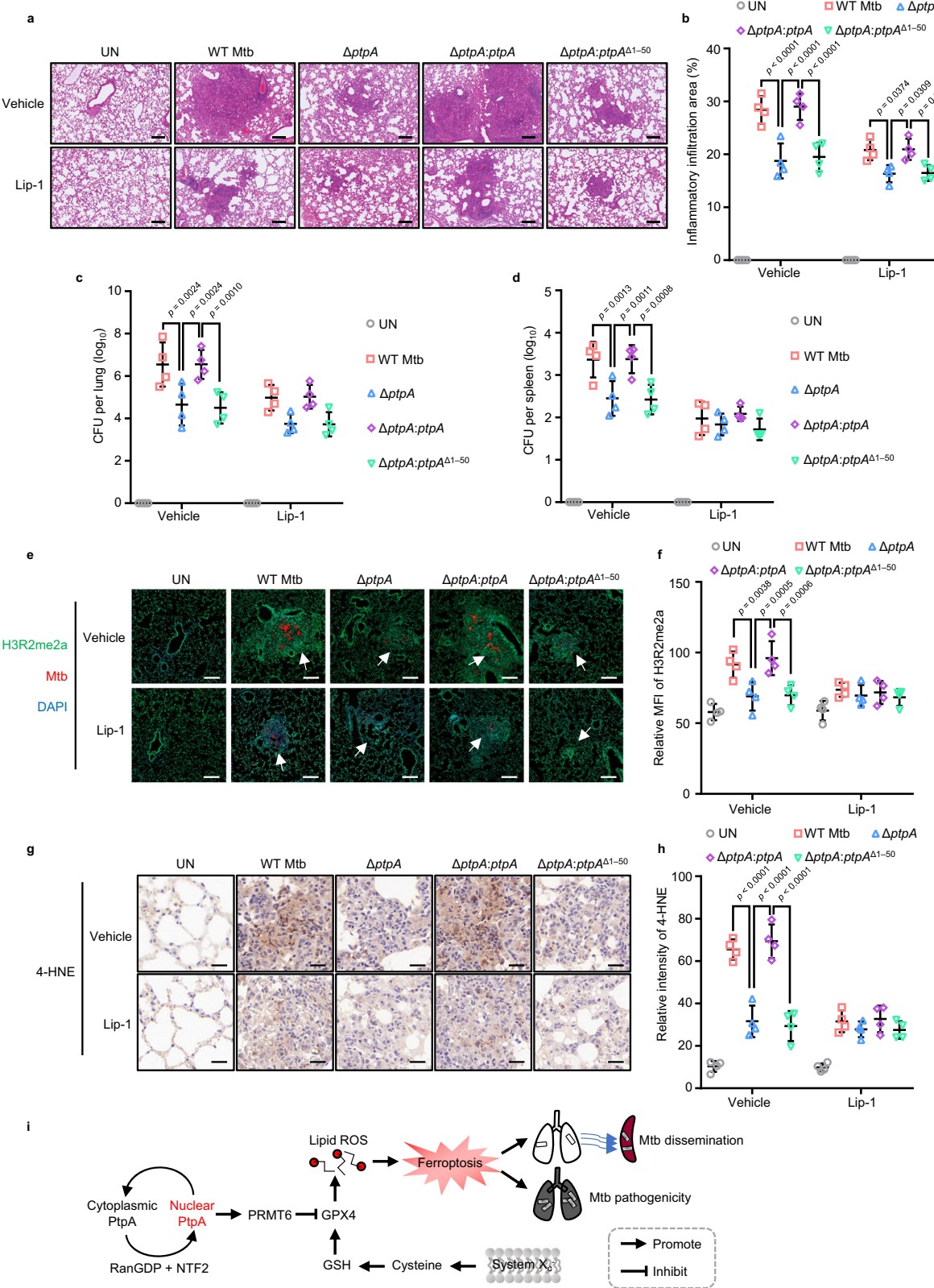

sepharose 4B resins in the binding buffer containing 50 mM Tris (pH 7.5), 150 mM NaCl, 5 mM DTT, and 0.1% NP-40. The resins were washed four times and subjected to immunoblot analysis as indicated.

### Immunofluorescence and confocal microscopy

Cells were seeded on cover glasses and transfected with the indicated plasmids. After 24 h, cells were fixed, permeabilized, and blocked at room temperature. Primary antibodies were then applied overnight. After washing three times with PBS, the fluorophore-conjugated secondary antibodies were applied for 1 h. Then, the coverslips were mounted onto glass slides using Antifade Mounting Medium with DAPI. Confocal images were taken with the Leica SP8 confocal microscope (Leica Microsystems) and analyzed by the Leica Application Suite Las X (v2.0.1.14392) software.

**Fig. 6 | Mtb PtpA induces ferroptosis to promote Mtb pathogenicity and dissemination in vivo. a** Hematoxylin and eosin (H&E) of lung sections from mice uninfected or infected with WT Mtb, Mtb Δ*ptpA*, Mtb Δ*ptpA:ptpA*, or Mtb Δ*ptpA:ptpA*$^{Δ1-50}$ strain for 5 weeks. Scale bars, 200 μm. Daily intraperitoneal injections of vehicle or Lip-1 (3 mg/kg) were administered to mice from day 15 after Mtb infection (*n* = 4). **b** Quantification of inflammatory areas in lungs from mice treated as in (**a**). **c** Mycobacterial survival in the lungs from mice treated as in (**a**). **d** Mycobacterial survival in the spleen from mice as in (**a**). **e** Immunofluorescence of H3R2me2a in lung sections from mice treated as in (**a**). Mtb strains (red) were stained with anti-Mtb antibody, H3R2me2a (Green) were stained with anti-

H3R2me2a antibody, and nuclei (blue) were stained with DAPI. Scale bars, 200 μm. **f** Quantification of the mean intensity of H3R2me2a in lung sections from mice treated as in (**e**). **g** Immunohistochemical staining of 4-HNE in lung sections from mice treated as in (**a**). Scale bars, 20 μm. **h** Quantification of 4-HNE expression in lung sections from mice treated as in (**g**). **i** Proposed model depicting Mtb PtpA-induced ferroptosis as well as pathogen pathogenicity and dissemination. Error bars are means ± SD of four mice per group and each represents data from two separate experiments. Statistical significance was determined using two-way ANOVA (Tukey's multiple comparisons test). Source data are provided as a Source Data file.

## Live-cell image

For analysis of dynamic cell viability, cells were plated at Glass Bottom Culture Dishes (NEST, 801002). After 24 h, cells were treated with 2 μM RSL3 and were then incubated with 5 μg/mL PI. Imaging was performed using Leica SP8 confocal microscope equipped with an environmental control chamber providing 37 °C, 5% $CO_2$, and 20–30% humidity. Images were acquired in 5 min intervals over a time frame of 6 h.

## RNA sequencing

Total RNA was extracted from the U937 cells infected with WT BCG or BCG Δ*ptpA* strain according to the manual of TRIzol® (Life Technologies, 15596-026). Sequencing was conducted using the Illumina HiSeq X Ten platform (Novogene Bioinformatics Technology Co., Ltd., Beijing, China) with paired-end 150 bp reads. The adaptors and low-quality bases were assessed using FASTQC (v0.11.3). Trimmed reads were then aligned to the human ensembl 79 (GRCh38.p2) genome using STAR (v2.4.2a). The genes differentially expressed between WT BCG and BCG Δ*ptpA*-infected cells with the fold change > 1.2 and the *p*-value <0.05 were analyzed by DESeq2 (v1.18.1) in R (v3.4.4). The results of RNA sequencing are available at the Gene Expression Omnibus (accession number: GSE199069).

## ChIP-qPCR and RT-qPCR

According to the manufacturer's instructions (Millipore, 17-409), about $1 \times 10^6$ infected cells were crosslinked for 10 min with 1% formaldehyde at room temperature. After washing twice with cold PBS, cells were lysed in SDS lysis buffer. Chromatin was then sonicated to an average fragment size of 200–1000 bp. For anti-H3R2me2a immunoprecipitation, the diluted chromatin was then incubated with the antibody against H3R2me2a overnight, and then incubated with protein A Agarose for 2 h. The complexes were washed and eluted in the buffer containing 1% SDS and 0.1 M NaHCO$_3$. Immunoprecipitated DNA fragments were then analyzed by qPCR. For (Quantitative Reverse Transcription PCR) RT-qPCR analysis, total RNA was extracted from the U937 cells infected with mycobacterial strains or treated with chemical solvents, and was then reverse-transcribed into cDNA with Hifair® II 1st strand cDNA synthesis kit. RT-qPCR was carried out on an ABI 7500 Real-Time PCR System.

## In vitro nuclear import assay

The U937 cells were treated with 40 μg/mL Digitonin for 10 min and washed twice with nuclear transport buffer containing 20 mM HEPES (pH 7.3), 110 mM KOAc, 5 mM NaOAc, 2 mM MgOAc, 2 mM DTT, 0.5 mM EGTA, and protease inhibitors. Then semipermeable cells were incubated in nuclear transport buffer containing 10 mg/mL U937 cytosol, an energy-regenerating mixture (1 mM ATP, 0.1 mM GTP, 5 mM creatine phosphate, and 20 U/mL creatine phosphokinase), GST-PtpA, or GST-PRMT6 at 30 °C for 30 min. Cells were fixed with 4% paraformaldehyde for 15 min and washed three times with PBS. As mentioned previously, cells were finally incubated with the indicated primary and secondary antibodies and then evaluated with Leica SP8 confocal microscope.

## Cell viability, cell death assay, and LDH assay

For cell viability, cells were seeded in 96-well plates and evaluated with a CCK-8 Cell Counting Kit (DOJINDO, CCK-8) according to the manufacturer's instructions. For cell death assays, 5 μg/mL PI was added to cells at 37 °C for 20 min and cells were imaged using the Leica fluorescence microscope. For the LDH assay, the supernatants from U937 cells infected with Mtb strains and LDH regents were incubated in a sterile 96-well plate. Fluorescence was measured at Infinite F200pro plate reader (Tecan) at 490 nm absorbance.

## In vitro methyltransferase assay

The coupled fluorescence-based assay system[57] was used to measure the PRMT6 methyltransferase activity. For 80 μL scale, 1 μg GST-PRMT6 and 1 μg GST-PtpA were separately or together mixed with 4 μg Bz-Arg-AMC, and incubated at room temperature for 30 min. Then 200 μM AdoMet was added to the reaction and incubated at 30 °C for 2 h. The reaction came to a halt at 95 °C for 5 min. Then 20 μL of the 1 mg/mL trypsin solution was added to 80 μL of the reaction mixture and transferred into a black flat-bottom 96-well plate. After 1 h, the fluorescence intensity ($\lambda_{ex}$ = 360 nm, $\lambda_{em}$ = 465 nm) was detected using the Infinite F200pro plate reader.

## Lipid peroxidation assay

For BODIPY (581/591) $C_{11}$ staining, differentiated U937 cells ($1 \times 10^5$ cells per well) were seeded in 12-well plates. The cells were incubated with 1.5 μM BODIPY (581/591) $C_{11}$ for 20 min at 37 °C, and were then washed and resuspended in 200 μL fresh Hanks' Balanced Salt solution, eventually analyzed using a BD LSRFortessa SORP flow cytometer with excitation at 488 nm. The data were analyzed using FlowJo (v10) Software.

## Measurement of phagosomal pH

Mtb strains were labeled with the pH-sensitive pHrodo probe (pHrodo Red succinimidyl ester) at 37 °C. After 1 h, the bacteria were washed three times with Middlebrook 7H9 medium containing 0.05% Tween-80, and were then labeled with 25 g/mL pH-insensitive pHrodo probe (Alexa Fluor 488 succinimidyl ester) at 37 °C for 1 h. The bacteria were then washed again with Middlebrook 7H9 medium containing 0.05% Tween-80. U937 cells were then incubated with the double-labeled Mtb strains at an MOI of 10. After 2 h, cells were washed three times with PBS to exclude non-internalized bacteria, and then cultured for additional 2 h at 37 °C. Cells were analyzed using a BD LSRFortessa SORP flow cytometer. The data were analyzed using FlowJo (v10) Software. According to a calibration curve, the mean fluorescence ratios of the pH-sensitive and pH-insensitive probes were utilized to quantify the phagosomal pH.

## Statistics & reproducibility

For in vitro study, investigators were not blinded to the sample identities during data collection since the readouts were quantitative and not prone to the subjective judgment of investigators. For in vivo study, mice experiments and statistical analysis were performed by independent researchers in a blinded manner. The quantified data with statistical analysis were performed using GraphPad Prism (v8.0) software. Unpaired two-sided Student's *t* test, one-way ANOVA or two-way

ANOVA analysis followed by multiple comparisons was used for statistical analyses. Statistical significance and *p* value were mentioned in the figure legends and were indicated in the figures respectively. At least three biological replicates were included. The data are presented as mean ± SD, and additional details about the statistical tests and numbers of samples are indicated in the corresponding figure legends.

## Reporting summary

Further information on research design is available in the Nature Portfolio Reporting Summary linked to this article.

## Data availability

The RNA-sequencing data generated in this study have been deposited in NCBI Gene Expression Omnibus database under accession number GSE199069 and these data have been publicly released. The amino acid sequence was analyzed in the SMART database (https://smart.embl.de/smart/show_motifs.pl). All the data supporting the findings of this study are available within the article and its Supplementary Information. Source data are provided with this paper.

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

## Acknowledgements

This work was supported by the National Key Research and Development Program of China (2022YFC2302900 to C.H.L. and J.W., 2021YFA1300200 to L.Z., C.H.L., and J.W.), the National Natural Science Foundation of China (81825014 to C.H.L., 31830003 to C.H.L., 81871616 to J.W., and 82022041 to J.W.), the Strategic Priority Research Program of the Chinese Academy of Sciences (XDB29020000 to C.H.L.), Youth Innovation Promotion Association CAS (Y2022036 to J.W.), and the CAS Project for Young Scientists in Basic Research (YSBR-010 to J.W.). We thank F. Shao (National Institute of Biological Sciences, Beijing) for providing plasmids, T. Zhao (Institute of Microbiology, Chinese Academy of Sciences, Beijing) for helping with the flow cytometry data generation and analysis, and X. Zhang (Institute of Microbiology, Chinese Academy of Sciences, Beijing) for helping with confocal microscopic analysis.

## Author contributions

C.H.L. and J.W. designed the experiments and supervised the project. C.H.L., L.Z., and J.W. carried out the project administration and funding acquisition. L.Q., Y.Z., ZeH.L., P.G., S.T., Q.C., M.Z., X.Z., and B.L. performed the functional analyses, animal, and cell biological experiments. Zhe.L. performed the bioinformatic analyses. C.H.L., J.W., L.Q., and Y.Z. wrote the manuscript. C.H.L., L.Z., and J.W. were responsible for writing-review & editing. Y.P. and L.Z. provided technical assistance.

## Competing interests

The authors declare no competing interests.
