## [Peer Review File · Nature Communications]

A mycobacterial effector promotes ferroptosis-dependent pathogenicity and disseminationREVIEWER COMMENTS

Reviewer #1 (Remarks to the Author):

This manuscript seeks to understand the *M. tuberculosis* effectors that influence host cell ferroptosis and the role of that process in *Mtb* infection. There is convincing data that cell death via ferroptosis is important for *Mtb* infection. Specifically, *Mtb* induces ferroptosis in infected cells, this can be prevented by the ferroptosis inhibitor Fer1, and in mice, Fer1 treatment reverses lung pathology and reduces *Mtb* bacterial loads by about 10 fold in lung and spleen (Amaral et al). These authors have studied the *Mtb* protein PtpA, a protein phosphatase, on innate immune activation of infected cells. In this paper these authors ask whether expression of PtpA can influence ferroptosis induced by RSL3, a ferroptosis activator. PtpA ameliorates RSL3 induced ferroptosis to some degree and the authors then go on to extensively characterize the function of PtpA in *Mtb* induced ferroptosis. The very extensive results show:

- 1) *Mtb* lacking *ptpA* induces less ferroptosis than wild type in U937 and PBMCs and this is genetically complemented. Δ *ptpA* infected cells express more Gpx4.
- 2) PtpA is found in both cytosolic and nuclear fractions of infected cells, interacts with the Ran protein, and this interaction is dependent on cysteine 11. The C11A mutant does not induce ferroptosis, but a phosphatase active site mutant does, indicating that the ferroptosis inducing activity of PtpA is independent of its phosphatase activity.
- 3) PtpA interacts directly with PRMT6 and H3 methylation is stimulated by WT *Mtb* but not the *ptpA* mutant. Domain truncations of PtpA indicate that the N terminus interacts with PRMT6 and the C terminus with H3. *Mtb* expressing PtpA lacking its N terminus behaves like the PtpA null, both in cellular infection and mouse infection.
- 4) A strain lacking the PRMT6 interacting N terminus does not induce ferroptosis in vivo

The model that emerges is that PtpA localizes to the nucleus and influences the transcriptional induction of ferroptosis through its effects on PRMT6 mediated epigenetic regulation. In general, the data supporting this model is strong and the paper is well done. I have some suggestions to support some weaknesses in the data:

- 1) One of the challenges of this system is the seemingly pleiotropic functions of this phosphatase, as the authors have shown in their prior papers. Thus, one question that arises is: what is the contribution of nuclear ferroptosis activity to the overall virulence function of this effector? To this end, the following questions arise:
 - a. The major evidence that the nuclear form is important are the two mutations (C11A and the 1-50 truncation). However, the 1-50 truncation is used to invoke the PRMT6 interaction, but won't this protein also fail to localize to the nucleus due to the loss of the Ran interacting site? In this case, can one dissect the effect of loss of nuclear localization from altered balance of nuclear vs cytoplasmic localization, where the protein has other activities? Are these mutants also defective for the cytosolic activities noted previously?
 - b. For the mouse experiments, one way to dissect the contribution of PtpA's ferroptosis activity in vivo is to use Fer1. This has been done in Amaral et al. The prediction would be that WT and *ptpA* ko would be equivalent when ferroptosis is inhibited if the major effect of PtpA is that pathway.
- 2) Although a technical point, it is critical for the interpretation of the *Mtb* mutant complementation studies. The in vivo phenotypes of the PtpA mutants could be due to loss of protein stability (instead of the loss of a specific protein interaction as hypothesized). The authors have a PtpA antibody, so protein expression in these strains should be verified, as well as nuclear localization (see point 1).
- 3) For many of the cellular infection assays in which the *ptpA* mutant is impaired for ferroptosis induction, there is a potential confounding issue of reduced bacterial load due to attenuation. For example, in 1C, the stained bacterial load looks lower. CFU data from infections should be presented to answer this issue. Is the phenotype of the mutant occurring at equivalent cellular infection levels or is the effect on ferroptosis due to lower bacterial loads from other functions of PtpA?

Other points;

- 1) I don't find the confocal images to be convincing in many places. For example, Fig 2D,3B, 4D

either because the background staining is very high or the localization is not clear. In addition, negative controls are not provided such as FLAG or GFP staining in a cell that carries no FLAG or GFP tagged protein, for example.

Textual issues

- 1) Line 47: it is an overstatement to say that N acetyl cysteine improves patient treatment outcomes.
- 2) Lines 73-75: this sentence is not clear.
- 3) Line 108: The causality of the PtpA-GPX4 stated cannot be inferred yet.
- 4) Line 321. State the strain of Mtb used.
- 5) Figure 6J: model. I'm not sure one can say that the pathway specifically promotes dissemination. The CFU decrement in the lungs is the same as in spleen.

Reviewer #2 (Remarks to the Author):

In this study, the authors show that infection of pathogen *Mycobacterium tuberculosis* (Mtb) induces ferroptosis in host cells and pathological damages in lung partly through its effector protein PrpA. Mechanistically, this study shows that PtpA can enter into nucleus in host cells, and interacts with PRMT6, enhances PRMT6's methyltransferase activity, promotes PRMT6-mediated H3R2me2a on the GPX4 promoter to suppress GPX4 transcription, and eventually promotes ferroptosis in host cells.

Overall, this is an interesting and novel finding with detailed mechanistic studies. The results provide important insights into the role of ferroptosis in Mtb. The scope is suitable for the broad readership at Nature Communications. This reviewer has several technical comments to further improve the manuscript.

Control is missing in a few data: they need to include NC (mock) control cells in Fig. 1 and related data (as included in Fig. 3i). In Fig. 4c-f, an empty vector-expressing control cell line should be added in these data.

In fig. 6 animal studies, they need to analyze the effect of ferroptosis inhibitor lipoxstatin-1 treatment on rescuing lung damages infected with WT mtb and delta ptpA mtb groups (NC included as a baseline control), to show how much pathological effect by mtb infection is caused by ferroptosis (and if so, whether lipoxstatin-1 can still have rescuing effect in delta ptpA mtb group).

Mtb infection should decrease GPX4 expression (Fig. 1i, but as pointed out above, this data lacks a MOCK control) and promotes cell death (which include ferroptosis). Based on the proposed model, the authors should test to restore GPX4 expression in WT Mtb-infected cells (to the level similar to that in MOCK/NC cells) and examine whether GPX4 restoration can suppress Mtb infection-induced cell death (and if so, compare GPX4 restoration-mediated cell death suppression with Fer-1 effect).

Fig. 2c, can authors comment why PrpA exhibits the same binding with Ran, GDP-Ran, and GTP-Ran (under conditions without NTF2), and why NTF2 overexpression decreases PrpA binding to Ran and GTP-Ran but seems to even increase binding to GDP-Ran?

Some of the writing is not optimal. Suggest the authors to send their manuscript to a professional editor for manuscript editing.

Point-by-point responses to referees' comments

RE: Manuscript (NCOMMS-22-29654) “A bacterial effector hijacks arginine methyltransferase PRMT6 to promote ferroptosis-dependent pathogen pathogenicity and dissemination” by Qiang *et al.*

Reviewers' comments:

Reviewer #1:

This manuscript seeks to understand the *M. tuberculosis* effectors that influence host cell ferroptosis and the role of that process in Mtb infection. There is convincing data that cell death via ferroptosis is important for Mtb infection. Specifically, Mtb induces ferroptosis in infected cells, this can be prevented by the ferroptosis inhibitor Fer1, and in mice, Fer1 treatment reverses lung pathology and reduces Mtb bacterial loads by about 10 fold in lung and spleen (Amaral et al). These authors have studied the Mtb protein PtpA, a protein phosphatase, on innate immune activation of infected cells. In this paper these authors ask whether expression of PtpA can influence ferroptosis induced by RSL3, a ferroptosis activator. PtpA ameliorates RSL3 induced ferroptosis to some degree and the authors then go on to extensively characterize the function of PtpA in Mtb induced ferroptosis. The very extensive results show: 1) Mtb lacking ptpA induces less ferroptosis than wild type in U937 and PBMCs and this is genetically complemented Δ ptpA infected cells express more Gpx4. 2) PtpA is found in both cytosolic and nuclear fractions of infected cells, interacts with the Ran protein, and this interaction is dependent on cysteine 11. The C11A mutant does not induce ferroptosis, but a phosphatase active site mutant does, indicating that the ferroptosis inducing activity of PtpA is independent of its phosphatase activity. 3) PtpA interacts directly

with PRMT6 and H3 methylation is stimulated by WT Mtb but not the ptpA mutant. Domain truncations of PtpA indicate that the N terminus interacts with PRMT6 and the C terminus with H3. Mtb expressing PtpA lacking its N terminus behaves like the PtpA null, both in cellular infection and mouse infection. 4) A strain lacking the PRMT6 interacting N terminus does not induce ferroptosis in vivo. The model that emerges is that PtpA localizes to the nucleus and influences the transcriptional induction of ferroptosis through its effects on PRMT6 mediated epigenetic regulation. In general, the data supporting this model is strong and the paper is well done. I have some suggestions to support some weaknesses in the data.

R: We thank the reviewer for the insightful comments and valuable suggestions. We have further validated our data with additional experiments and revised our manuscript accordingly. The following is the point-by-point response to all comments.

Specific comments:

1. One of the challenges of this system is the seemingly pleiotropic functions of this phosphatase, as the authors have shown in their prior papers. Thus, one question that arises is: what is the contribution of nuclear ferroptosis activity to the overall virulence function of this effector? To this end, the following questions arise:

a) The major evidence that the nuclear form is important are the two mutations (C11A and the 1-50 truncation). However, the 1-50 truncation is used to invoke the PRMT6 interaction, but won't this protein also fail to localize to the nucleus due to the loss of the Ran interacting site? In this case, can one dissect the effect of loss of nuclear localization from altered balance of nuclear vs cytoplasmic localization, where the

protein has other activities? Are these mutants also defective for the cytosolic activities noted previously?

b) For the mouse experiments, one way to dissect the contribution of PtpA's ferroptosis activity *in vivo* is to use Fer1. This has been done in Amaral *et al.* The prediction would be that WT and ptpA ko would be equivalent when ferroptosis is inhibited if the major effect of PtpA is that pathway.

R: We thank the reviewer for raising this important question. Indeed, Mtb PtpA is a pleiotropic effector relying on distinct regions and/or enzymatic activities to regulate different pathogenic effects. Specifically, depending on its tyrosine phosphatase, PtpA functions as an important effector in suppressing phagosome acidification and immunological signaling pathways for immune evasion (*Cell Host Microbe*, 2008; *Nat Immunol*, 2015). And here we further reveal an additional function of nuclear PtpA in triggering ferroptosis to promote Mtb dissemination and tissue damage, which is independent of its tyrosine phosphatase activity.

a) Since we found that wild-type (WT) PtpA enters into host cell nuclei via its RanGDP-binding site Cys11 and then interacts with nuclear PRMT6 (via the 1-50 amino acid residues of PtpA) to trigger ferroptosis. We thus constructed PtpA^{C11A} and PtpA^{Δ1-50} mutants to examine their host cell nuclear entry, and found that PtpA^{C11A} failed to localize to the nucleus of host cells due to the loss of its RanGDP-interacting site. However, PtpA^{Δ1-50}, which also lacks the RanGDP-interacting site, retained its ability to localize to host cell nuclei (revised Fig. 5f), suggesting that PtpA^{Δ1-50} enters into host cell nuclei in a RanGDP-independent pathway. We then further explored the mechanism by which PtpA^{Δ1-50} mutant, a truncated mutant with a lower molecular weight of 12 KDa, enters into the nucleus of host cells by conducting an *in vitro* nuclear import assay, and we found that PtpA^{Δ1-50} had a

noticeable shift to the nuclei in the absence of the nuclear transport complex comprising RanGDP and NTF2 (revised Supplementary Fig. 6c). Thus, these results suggested that PtpA^{Δ1-50}, with a lower molecular weight, can freely diffuse into the host cell nuclei, independent of the RanGDP/NTF2 complex-mediated nuclear import system. Our data are consistent with a previous report demonstrating that the central structure of the nuclear pore complex contains a central channel that allows the transport of proteins with a molecular mass of less than 40 kDa by passive diffusion (*Dev Cell*, 2003). Taken together, we thus speculated that PtpA^{Δ1-50} might enter into the host cell nuclei by passive diffusion.

Previous studies demonstrated that cytoplasmic PtpA inhibits host innate immunity (including phagosome acidification and JNK/p38 MAPK signaling pathways) in a tyrosine phosphatase-dependent manner (*Cell Host Microbe*, 2008; *Nat Immunol*, 2015). We then further examined whether PtpA^{C11A} and PtpA^{Δ1-50} have cytosolic activities, and found that both PtpA^{C11A} and PtpA^{Δ1-50} lost their phosphatase activity, thus failing to inhibit phagosome acidification and JNK/p38 MAPK signaling pathway activation in host cells (revised Supplementary Fig. 6f-h). In addition, we determined that the cytosolic activity of PtpA did not affect ferroptosis. Specifically, cell viability assay revealed that PtpA^{C11A} and PtpA^{Δ1-50} lost their capacity to trigger ferroptosis, whereas the phosphatase-dead PtpA^{D126A} mutant, which retains the nuclear entry ability, still had the activity to induce ferroptosis (revised Fig. 2g, h and revised Fig. 5e, f). Taken together, our data indicate that PtpA triggers ferroptosis depending on its nuclear activity through the RanGDP-binding site (Cys11) and PRMT6-binding region (1-50 amino acids), independent of its tyrosine phosphatase-dependent cytosolic activity.

b) We thank the reviewer for this valuable suggestion. We have repeated the

mouse experiments in which the ferroptosis inhibitors (including two potent lipid peroxidation inhibitors Fer-1 and Lip-1) treatment groups were included (revised Fig. 6). During Mtb infection, WT Mtb-infected mice showed more severe pathological damage and elevated lipid peroxidation level in lungs, as well as higher mycobacterial load in lungs and spleens, compared with Mtb $\Delta ptpA$ strain-infected mice, while ferroptosis inhibitors could significantly diminish the above differences between WT Mtb- and Mtb $\Delta ptpA$ strain-infected mice. In fact, WT Mtb-infected mice still showed slightly higher pathological damage and mycobacterial load in lungs compared with Mtb $\Delta ptpA$ strain-infected mice after ferroptosis inhibitor treatment, probably due to the fact that PtpA promotes Mtb pathogenicity and intracellular survival partially depending on its cytosolic activity by suppressing host phagosome acidification and JNK/p38 MAPK signaling pathway activation (*Nat Immunol*, 2015).

2. Although a technical point, it is critical for the interpretation of the Mtb mutant complementation studies. The *in vivo* phenotypes of the PtpA mutants could be due to loss of protein stability (instead of the loss of a specific protein interaction as hypothesized). The authors have a PtpA antibody, so protein expression in these strains should be verified, as well as nuclear localization see point 1).

R: We thank the reviewer for this constructive suggestion. We have performed immunoblot analysis and confocal microscopic analysis to detect protein stability and localization of PtpA and its mutants using PtpA antibody. We found that stably expressed PtpA and its mutants could be secreted into infected U937 cells with comparable protein levels (revised Supplementary Fig. 6d, e, h). In addition, we confirmed that PtpA ^{$\Delta 1-50$} was located in both the cytoplasm and nucleus of host

cells, whereas PtpA^{C11A} was only located in the cytoplasm of host cells during Mtb infection (revised Supplementary Fig. 6d, e).

3. For many of the cellular infection assays in which the ptpA mutant is impaired for ferroptosis induction, there is a potential confounding issue of reduced bacterial load due to attenuation. For example, in 1C, the stained bacterial load looks lower. CFU data from infections should be presented to answer this issue. Is the phenotype of the mutant occurring at equivalent cellular infection levels or is the effect on ferroptosis due to lower bacterial loads from other functions of PtpA?

R: We thank the reviewer for raising this concern. Previous studies and our data have indicated that Mtb PtpA could promote Mtb intracellular survival in host cells (*Cell Host Microbe*, 2008; *Nat Immunol*, 2015) (revised Supplementary Fig. 2b). To confirm that the functional deficiency of PtpA mutants, rather than the lower bacteria load, is the cause of impaired ferroptosis, we further performed bacterial colony-forming units (CFUs) assay in which additional Mtb mutants including $\Delta ptpA:ptpA^{D126A}$ (which loses its phosphatase activity but retains its nuclear entry ability) and Mtb $\Delta ptpA:ptpA^{C11A}$ (which loses its phosphatase activity and nuclear entry ability) strains were included. As shown in the revised Supplementary Fig. 3d, the mutant strains including Mtb $\Delta ptpA$, Mtb $\Delta ptpA:ptpA^{D126A}$, and Mtb $\Delta ptpA:ptpA^{C11A}$ all had a significant decrease in Mtb survival in U937 cells as compared with that of WT Mtb strain and the complemented Mtb $\Delta ptpA:ptpA$ strain. Meanwhile, cell viability data showed that WT Mtb, Mtb $\Delta ptpA:ptpA$, and Mtb $\Delta ptpA:ptpA^{D126A}$ strains, but not Mtb $\Delta ptpA$ and Mtb $\Delta ptpA:ptpA^{C11A}$ strains, significantly promoted ferroptosis of U937 cells

(revised Fig. 2g, h). Thus, these results indicate that Mtb PtpA-induced ferroptosis mainly depends on its nuclear activity, rather than the bacterial load.

Other points:

1. I don't find the confocal images to be convincing in many places. For example, Fig 2D, 3B, 4D either because the background staining is very high or the localization is not clear. In addition, negative controls are not provided such as FLAG or GFP staining in a cell that carries no FLAG or GFP tagged protein, for example.

R: We thank the reviewer for raising these concerns. We have repeated the relevant experiments and tried to optimize the experimental conditions to improve data quality, in which empty vectors-expressing cells and uninfected cells were added as the negative control (revised Fig. 1c-f, 1h-j, 2a, 2c-e, 2g, 2h, 3b, 3c,4c-f, 5g-j and Supplementary Fig. 3a).

Textual issues:

1. Line 47: it is an overstatement to say that N acetyl cysteine improves patient treatment outcomes.

R: We thank the reviewer for reminding us of this issue. We have corrected this sentence in the revised manuscript.

2. Lines 73-75: this sentence is not clear.

R: We thank the reviewer for reminding us of this issue. We have rephrased this sentence in the revised manuscript.

3. Line 108: The causality of the PtpA-GPX4 stated cannot be inferred yet.

R: We thank the reviewer for pointing out this issue. We have rephrased this sentence in the revised manuscript.

4. Line 321. State the strain of Mtb used.

R: We thank the reviewer for this good suggestion. We have stated the strain of Mtb used in this study in the “Methods” section of the revised manuscript.

5. Figure 6J: model. I’m not sure one can say that the pathway specifically promotes dissemination. The CFU decrement in the lungs is the same as in spleen.

R: We thank the reviewer for raising this concern. Previous studies have indicated that ferroptosis could promote both Mtb pathogenicity and dissemination (*J Exp Med*, 2019; *J Exp Med*, 2022). Consistently, as shown in the revised Fig. 6, Mtb PtpA induces ferroptosis and promotes Mtb pathogenicity as well as Mtb dissemination in Mtb-infected mice. Accordingly, we have modified the model to improve comprehension (revised Fig. 6i).

Reviewer #2:

In this study, the authors show that infection of pathogen *Mycobacterium tuberculosis* (Mtb) induces ferroptosis in host cells and pathological damages in lung partly through its effector protein PtpA. Mechanistically, this study shows that PtpA can enter into nucleus in host cells, and interacts with PRMT6, enhances PRMT6’s methyltransferase activity, promotes PRMT6-mediated H3R2me2a on the GPX4 promoter to suppress GPX4 transcription, and eventually promotes ferroptosis in host cells. Overall, this is an interesting and novel finding with detailed mechanistic studies. The results provide important insights into the role of ferroptosis in Mtb. The scope is suitable for the broad

readership at Nature Communications. This reviewer has several technical comments to further improve the manuscript.

R: We thank the reviewer for the concise summary and encouraging comments.

Specific comments:

1. Control is missing in a few data: they need to include NC (mock) control cells in Fig. 1 and related data (as included in Fig. 3i). In Fig. 4c-f, an empty vector-expressing control cell line should be added in these data.

R: We thank the reviewer for reminding us of these issues. We have repeated the relevant experiments, in which empty vectors-expressing cells and uninfected cells were added as the control (revised Fig. 1c-f, 1h-j, 2a, 2c-e, 2g, 2h, 4c-f, 5g-j and Supplementary Fig. 3a).

2. In fig. 6 animal studies, they need to analyze the effect of ferroptosis inhibitor lipoxstatin-1 treatment on rescuing lung damages infected with WT mtb and delta ptpA mtb groups (NC included as a baseline control), to show how much pathological effect by mtb infection is caused by ferroptosis (and if so, whether lipoxstatin-1 can still have rescuing effect in delta ptpA mtb group).

R: We thank the reviewer for this constructive suggestion. We have repeated the mouse experiments in which the ferroptosis inhibitors (including Fer-1 and Lip-1) treatment groups and uninfected control groups were added (revised Fig. 6). Our data showed that during Mtb infection, WT Mtb-infected mice exhibited more severe pathological damage and elevated lipid peroxidation level in lungs as well as higher bacterial load in lungs and spleens compared with Mtb Δ ptpA-infected mice, while ferroptosis inhibitors could significantly diminish the above

differences. In these experiments, ferroptosis inhibitors could rescue ferroptosis-driving tuberculosis pathology, whereas WT Mtb-infected mice still showed slightly higher pathological damage and bacterial load in lungs compared with Mtb $\Delta ptpA$ -infected mice after ferroptosis inhibitors treatment, which is probably due to the facts that PtpA promotes Mtb pathogenicity and intracellular survival partially depending on its cytosolic activity by suppressing host phagosome acidification and JNK/p38 MAPK signaling pathway activation (*Nat Immunol*, 2015).

3. Mtb infection should decrease GPX4 expression (Fig. 1i, but as pointed out above, this data lacks a MOCK control) and promotes cell death (which include ferroptosis). Based on the proposed model, the authors should test to restore GPX4 expression in WT Mtb-infected cells (to the level similar to that in MOCK/NC cells) and examine whether GPX4 restoration can suppress Mtb infection-induced cell death (and if so, compare GPX4 restoration-mediated cell death suppression with Fer-1 effect).

R: We thank the reviewer for this valuable suggestion. We have adopted the Tet-On inducible gene expression system to restore *GPX4* gene expression by doxycycline (DOX) in U937 cells. As shown by immunoblot analysis, we have confirmed that DOX could indeed restore GPX4 expression in cells infected with WT Mtb or Mtb $\Delta ptpA:ptpA$ strain to the similar level as that in uninfected cells without DOX treatment (revised Supplementary Fig. 2i). Then, we analyzed the lipid peroxidation levels and cell viability of U937 cells, and found that GPX4 restoration can suppress Mtb-induced ferroptosis (revised Fig. 1i, j). It should be mentioned that we also noticed that Fer-1 (a potent inhibitor of lipid peroxidation) treatment was more effective than GPX4 restoration in suppressing Mtb-induced

ferroptosis. Since multiple signaling pathways, including cysteine-GSH-GPX4 axis, iron metabolism, reactive oxygen species (ROS) metabolism, and MAPK pathway, could promote the production of lipid peroxidation to induce ferroptosis (*Cell Death Differ*, 2016; *Cell*, 2017), we thus speculated that besides Mtb PtpA-GPX4 axis-mediated ferroptosis activation, Mtb might regulate host ferroptosis partially by targeting additional pathways, such as iron metabolism, ROS metabolism, and MAPK pathway.

4. Fig. 2c, can authors comment why PtpA exhibits the same binding with Ran, GDP-Ran, and GTP-Ran (under conditions without NTF2), and why NTF2 overexpression decreases PtpA binding to Ran and GTP-Ran but seems to even increase binding to GDP-Ran?

R: We thank the reviewer for pointing out these issues. We have repeated the pull-down experiment with the optimized experimental conditions and confirmed that PtpA exhibits the same binding capacity with Ran, RanGDP, and RanGTP under conditions without NTF2 (revised Fig. 2c), suggesting that the binding motif of PtpA among Ran, RanGDP, and RanGTP are conserved without overlapping with the GTP hydrolysis and nucleotide binding site in Ran. Moreover, NTF2 increases the interaction of PtpA with RanGDP, but not with Ran or RanGTP. Our data are consistent with a previous report demonstrating that the nuclear import factor NTF2 can directly interact with RanGDP (but not Ran or RanGTP) to induce a major conformational change in RanGDP to promote its interaction with the cargo proteins, such as ankyrin repeats-containing proteins (*Cell*, 2014).

5. Some of the writing is not optimal. Suggest the authors to send their manuscript to a professional editor for manuscript editing.

R: We thank the reviewer for the kind suggestion. We have sent the manuscript to a professional editor for manuscript editing.

Once again, we greatly appreciate the reviewers for having helped us improve this manuscript tremendously.

REVIEWERS' COMMENTS

Reviewer #1 (Remarks to the Author):

I thank the authors for responding to the prior critique. I find the revisions satisfactory in answering the points raised.

Reviewer #2 (Remarks to the Author):

The authors have adequately addressed the comments from this reviewer. Overall, this is a very nice study. I therefore recommend its publication in Nature Communications.

One comment, in the ferroptosis field, it is generally accepted that the ferroptosis inhibitor ferrostatin-1 is a poor drug for in vivo treatment (most studies use liproxstatin-1 as a ferroptosis inhibitor for in vivo treatment). To avoid confusion in the field, I suggest the authors to remove their ferrostatin-1 in vivo treatment data in Fig. 6 (since they have liproxstatin-1 in vivo data).

Point-by-point responses to referees' comments

RE: Manuscript (NCOMMS-22-29654A) “**A mycobacterial effector promotes ferroptosis-dependent pathogenicity and dissemination**” by Qiang *et al.*

Reviewers' comments:

Reviewer #1:

I thank the authors for responding to the prior critique. I find the revisions satisfactory in answering the points raised.

R: We appreciate the reviewer's positive remarks on our revised paper.

Reviewer #2:

The authors have adequately addressed the comments from this reviewer. Overall, this is a very nice study. I therefore recommend its publication in Nature Communications. One comment, in the ferroptosis field, it is generally accepted that the ferroptosis inhibitor ferrostatin-1 is a poor drug for in vivo treatment (most studies use liproxstatin-1 as a ferroptosis inhibitor for in vivo treatment). To avoid confusion in the field, I suggest the authors to remove their ferrostatin-1 in vivo treatment data in Fig. 6 (since they have liproxstatin-1 in vivo data).

R: We thank the reviewer for the valuable suggestion. We have removed the ferrostatin-1 in vivo treatment data in revised Fig. 6 and Supplementary Fig. 7, and revised our manuscript and Supplementary Information files accordingly.

Once again, we greatly appreciate the reviewers for helping us improve this manuscript tremendously.